# Dissociation between individual differences in self-reported pain intensity and underlying fMRI brain activation

M. E. Hoeppli [1,2 ✉], H. Nahman-Averbuch[1,2,3], W. A. Hinkle[1], E. Leon [1,2], J. Peugh[2,4], M. Lopez-Sola[5], C. D. King [1,2,4], K. R. Goldschneider[1,6] & R. C. Coghill [1,2,4]

Pain is an individual experience. Previous studies have highlighted changes in brain activation and morphology associated with within- and interindividual pain perception. In this study we sought to characterize brain mechanisms associated with between-individual differences in pain in a sample of healthy adolescent and adult participants (N = 101). Here we show that pain ratings varied widely across individuals and that individuals reported changes in pain evoked by small differences in stimulus intensity in a manner congruent with their pain sensitivity, further supporting the utility of subjective reporting as a measure of the true individual experience. Furthermore, brain activation related to interindividual differences in pain was not detected, despite clear sensitivity of the Blood Oxygenation Level-Dependent (BOLD) signal to small differences in noxious stimulus intensities within individuals. These findings suggest fMRI may not be a useful objective measure to infer reported pain intensity.

[1] Pediatric Pain Research Center (PPRC), Cincinnati Children's Hospital Medical Center, Cincinnati, OH, USA. [2] Division of Behavioral Medicine and Clinical Psychology, Cincinnati Children's Hospital Medical Center, Cincinnati, OH, USA. [3] Division of Clinical and Translational Research and Washington University Pain Center, Department of Anesthesiology, Washington University School of Medicine, St Louis, MO, USA. [4] Department of Pediatrics, University of Cincinnati, College of Medicine, Cincinnati, OH, USA. [5] Serra Hunter Programme, Department of Medicine, School of Medicine and Health Sciences, University of Barcelona, Barcelona, Spain. [6] Pain Management Center, Department of Anesthesiology, Cincinnati Children's Hospital Medical Center, Cincinnati, OH, USA. ✉email: marie-eve.hoeppli@cchmc.org

nterindividual differences in the experience of pain can be profound and represent a major clinical challenge. Even when pain is evoked by carefully controlled experimental stimulus of a fixed intensity, pain intensity ratings range extensively across healthy individuals[1–3]. Moreover, activation of the primary somatosensory cortex (SI), the anterior cingulate cortex (ACC), and the prefrontal cortex (PFC) has been shown to be related to interindividual differences in the experience of pain evoked by a noxious stimulus[1,3–8]. Similarly, these areas have been associated with within-individual differences in pain using fixed noxious stimuli of graded intensity[9–12]. These areas have been previously and consistently associated with nociceptive processing and pain[13–18]. Similarly, gray matter densities have been shown to be associated with interindividual differences in pain sensitivity in areas frequently associated with pain processing[19,20]. Extending these observations, machine-learning techniques have been used to develop a multi-voxel brain signature of pain that is sensitive to small within-individual differences in pain and is specific to physical pain compared to other aversive states, such as social pain and anticipation of pain, which produce no expression of such signatures[21–24]. Taken together, these studies would strongly suggest that neuroimaging data could provide an objective biomarker for pain[25].

The search for an objective biomarker for pain intensity has been driven by a legitimate need to adequately assess pain in individuals who are unable to adequately communicate their first-person experience of pain to a third person[26], as well as to understand changes in such markers over time in response to disease or treatment[27–30]. However, there is also substantial pressure to use such markers to confirm the veracity of a patient's report of pain for legal and financial reasons[26,27,30,31].

A critical criterion for biomarkers for individual pain intensity is that they are sensitive to individual differences in reported pain magnitude such that they could accurately distinguish between individuals experiencing a great deal of pain vs. individuals experiencing relatively low levels of pain. In order for such biomarkers to be developed using neuroimaging techniques, the feasibility of detecting brain activation related to interindividual differences in reported pain using fMRI needs to be confirmed. Although previous studies have already highlighted interindividual differences in the experience of pain and potential underlying brain mechanisms, these studies often relied on very small sample sizes, including 25 participants or fewer, and narrow age ranges, including mostly young adults in their twenties[1,3,4]. This limits the ability to generalize to the general population and confirm by replication relationships between brain regions and reported pain[32,33].

In the present investigation, we aimed to further characterize brain mechanisms that support interindividual differences in the experience of pain in a sample of over 100 participants. Defining such mechanisms using carefully controlled noxious stimuli in healthy individuals is an essential first step to developing valid markers of interindividual differences in pain intensity as such stimuli provide an objective input into the nociceptive system. To address the issues of previous studies examining individual differences in pain, we included more than 100 participants and adolescents, as well as adults ranging into the 5th decade, in addition to young adults. In addition, to be more representative of the general population, enrollment criteria defined for this study did not exclude individuals based on their pain sensitivity or individuals with anxiety or depression.

The characterization of individual differences in pain relies heavily on the accurate communication of the first-person subjective pain experience. Accordingly, we used well-validated visual analog scales (VAS) for rating pain intensity and pain unpleasantness[34,35].

To better determine if ratings of pain intensity and pain unpleasantness reflected interindividual differences in the magnitude of the subjective experience of pain, we first performed a Gaussian Mixture Model (GMM) analysis on responses to graded noxious heat stimuli ranging from 35 °C to 49 °C. To describe the pain intensity and unpleasantness responses of each pain sensitivity class, we fitted the data to a power function in order to extract the exponent and proportionality constant.

To rule out the possibility that apparent pain sensitivity reflected a systematic bias in the reporting of the subjective experience vs. true interindividual differences in the actual experience of pain, we examined the psychological profiles of individuals within each class.

To further confirm that the VAS ratings of pain reflected actual pain sensitivity rather than reporting biases, we examined the ability of participants within each pain sensitivity class to discriminate small differences in noxious stimulus intensity.

To characterize the relationship between brain activation and interindividual differences in perceived intensity, we examined brain responses to both noxious and innocuous stimulation using BOLD fMRI. To replicate and extend our previous findings[1], we focused primarily on responses to noxious heat. We further investigated responses to noxious cold to determine if findings from noxious heat would generalize to another painful modality. Finally, auditory stimulation was utilized to assess the ability of fMRI to detect individual differences in perceived stimulus intensity in a non-noxious sensory modality. Using a GLM-based massive univariate analysis, we tested the hypothesis that interindividual differences in reported painful and non-painful sensations would be associated with interindividual differences in brain activation. To ensure that our neuroimaging methods were sufficiently sensitive to detect brain activation associated with small differences in stimulus intensity, we examined within-individual responses to two levels of heat (47 °C, 48 °C), cold (0.5 °C, 3 °C), and auditory stimuli (80 dB, 90 dB). The results of the massive univariate analyses were confirmed with additional analyses using the Neurologic Pain Signature[21] and a multivariate LASSO-PCR (least absolute shrinkage and selection operator-regularized principal components regression) regression[36].

Our results show that individuals perceive and report pain sensations associated with different intensity of noxious stimuli in a manner congruent with their pain sensitivity. In addition, our results show that interindividual differences in the perception of pain intensity are not associated with changes in brain activation. Taken together, these results support the validity of pain ratings as a measure to assess one's pain sensations, while highlighting the need to consider fMRI brain imaging an objective measure of pain sensations very cautiously.

## Results

Across all noxious stimuli delivered in both the QST session and the MRI session, tremendous interindividual differences in pain sensitivity were noted, with VAS pain intensity ratings to 48 °C stimuli ranging from nearly 0 to 10 (max). In order to better understand these interindividual differences, we performed a series of analyses on the psychophysical data.

**Heat pain sensitivity classes.** A mixture model analysis was performed on pain intensity and pain unpleasantness ratings of graded noxious heat stimuli to define the number of pain sensitivity classes and assign participants to these classes. Two criteria were used to select the best fitting model for the number of pain classes: no spurious class and lowest BIC or Bayesian Information Criterion value. Given that the model for four and five classes did not meet the selection criteria by creating at least one spurious

class, these models were eliminated and only the models for one through three classes were considered in the selection of the best fitting model. BIC values resulting from the mixture models analysis ranged from 23518.905 for the 'one class' model to 19791.321 for the 'three classes' model. As a result, the model with three classes was chosen because it had the lowest BIC value. The three classes of pain sensitivity were defined as Low, Moderate, and High Pain Sensitivity. Each class of pain sensitivity had a very distinctive power function for the pain intensity and unpleasantness ratings, as shown by their respective exponents and proportionality constants. Participants from the High Pain Sensitivity class showed the smallest exponents (intensity: exponent = 2.93; unpleasantness: exponent = 3.45) but had the highest proportionality constants (intensity: constant = −5.49; unpleasantness: constant = −6.88) in reported pain intensity and unpleasantness (Supplementary Fig. 1C). These coefficients are likely explained by the combination of relatively high ratings in response to stimuli of low noxious intensity (43°) as well as a flattening of the ratings at the high end of the noxious range (48–49 °C). The fit of intensity and unpleasantness ratings to the power function for this class were both significant (intensity: $F_{(1,5)} = 328.1$, $p < 0.0001$; unpleasantness: $F_{(1,5)} = 424$, $p < 0.0001$). The individuals from the Low Pain Sensitivity class had greater exponents (intensity: exponent = 3.88; unpleasantness: exponent = 4.74) and low constants (intensity: constant = −9.79; unpleasantness: constant = −12.1) (Supplementary Fig. 1A) than participants of the High Pain Sensitivity class. The fit of intensity and unpleasantness ratings to the power function for this class were significant (intensity: $F_{(1,5)} = 139.7$, $p < 0.0001$; unpleasantness: $F_{(1,5)} = 134.5$, $p < 0.0001$). Similarly, the participants from the Moderate Pain Sensitivity class had greater exponents (intensity: exponent = 4.46; unpleasantness: exponent = 5.12) and low constants (intensity: constant = −10.02; unpleasantness: constant = −11.84) than participants of the High Pain Sensitivity class (Supplementary Fig. 1B). The fit of intensity and unpleasantness ratings to the power function for this class were significant (intensity: $F_{(1,5)} = 508.9$, $p < 0.0001$; unpleasantness: $F_{(1,5)} = 413.1$, $p < 0.0001$). Furthermore, exponents of unpleasantness ratings were always greater than those of intensity ratings and their proportionality constants were always lower.

Finally, pain sensitivity classes did not differ in demographic factors, such as sex, age, race, economic status, or handedness, nor in psychological factors, including measures of sleeping patterns (scores to the Epworth Sleepiness Scale and Pittsburgh Sleep Quality Index), measures of emotional states (scores to PROMIS anxiety, depression, and pain interference, PANAS, Barratt Impulsiveness Scale, Freiburg Mindfulness Scale, and Pain Catastrophizing Scale) or in scores to the Experience of Discrimination scale (Supplementary Table 1).

**Rating-based Discrimination thresholds**. In order to further confirm that reports of pain largely reflected the subjective experience rather than a rating bias, we examined the ability of participants to provide ratings that discriminated between small differences in stimulus intensity. Kruskal–Wallis tests performed on the 43 °C-ascending discrimination thresholds revealed a significant difference between classes in their ability to discriminate using intensity and unpleasantness ratings (intensity: $\chi^2(2) = 10.627$, $p = 0.005$; unpleasantness: $\chi^2(2) = 16.425$, $p = 0.0003$). Post-hoc Dunn tests on rating-based discrimination thresholds of intensity ratings showed significant differences between classes of Low and Moderate Pain Sensitivity ($z = −2.53$, adjusted $p = 0.017$) and between classes of Low and High Pain Sensitivity ($z = 2.97$, adjusted $p = 0.009$) (Fig. 1A), such that

participants from the Low Pain Sensitivity class required larger temperature steps to reliably perceive stimuli as more intense than 43 °C. Similarly, post-hoc Dunn tests on rating-based discrimination thresholds in unpleasantness ratings showed significant differences between classes of Low and Moderate Pain Sensitivity ($z = −3.05$, adjusted $p = 0.003$) and between classes of Low and High Pain Sensitivity ($z = 3.75$, adjusted $p = 0.0005$), again with larger steps required before participants from the Low Pain Sensitivity class experienced reliably greater pain unpleasantness.

Kruskal–Wallis tests performed on the 49 °C-descending discrimination thresholds revealed no significant difference between classes in intensity ratings ($\chi^2(2) = 0.98063$, $p = 0.6$). However, they detected a significant difference between classes in unpleasantness ratings ($\chi^2(2) = 13.625$, $p = 0.001$). Post-hoc Dunn tests showed significant differences between classes of Moderate and High Pain Sensitivity ($z = 3.32$, adjusted $p = 0.001$) and between classes of Low and High Pain Sensitivity ($z = 3.33$, adjusted $p = 0.003$) (Fig. 1B). In this case, larger temperature decreases were needed for the individuals from the High Pain Sensitivity class to feel less pain unpleasantness than other classes.

**Relationship of reported heat pain intensity with brain activation**. Heat pain intensity ratings of high noxious stimuli, i.e., 48 °C, during the fMRI series ranged widely across individuals from 0.07 to 10 with an average rating of 3.92 ± 2.63 on the VAS (Fig. 2A and Supplementary Fig. 2). These interindividual differences provide a wide range of ratings to assess the relationship between individual differences in reported pain intensity and brain activation associated with high intensity stimuli (48 °C). To investigate this relationship, a GLM analysis was performed examining the main effect of stimulation with individual pain intensity ratings as the covariate of interest. Analyses of brain activation in response to high intensity heat stimulus (48 °C) (Fig. 2B, Supplementary Tables 2 and 3) revealed increased activation in areas such as the cerebellum, putamen, caudate nucleus, thalamus, primary and secondary somatosensory cortices (SI; SII), insula, the anterior cingulate cortex (ACC), and the dorsolateral prefrontal cortex (DLPFC). Decreased activation was observed in areas such as the amygdala and hippocampus, as well as the posterior cingulate cortex (PCC) and the precuneus.

Surprisingly, results of the interindividual covariance analysis detected no relationship between individually reported pain intensity and increased or decreased brain activation associated with high intensity heat stimuli at a clustering z threshold of 3.1 and a p threshold of 0.05 (Fig. 2C).

To investigate the lack of relationship between brain activation associated with high intensity heat stimulus and reported pain intensity and ensure that it was not due to the statistical approach, i.e., univariate analysis, the NPS was applied to the heat fMRI data and correlated with the individual pain intensity ratings. Results of this analysis confirmed that all participants exhibited positive expression of the NPS during the high intensity heat stimulus (Supplementary Fig. 3A). Results of the interindividual correlation analysis between the NPS expression and reported pain intensity failed to show a significant association: R (99) = 0.002, p > 0.8 (Supplementary Fig. 3B).

To further ensure that the lack of relationship between brain activation associated with high intensity heat stimuli and perceived pain intensity was not due to a lack of sensitivity of the univariate approach or to the fact that the NPS was not trained on our data, we performed a multivariate LASSO-PCR regression and trained it on our dataset. Results of this analysis showed no relationship between individual differences in pain intensity ratings and brain activation, further supporting our main finding

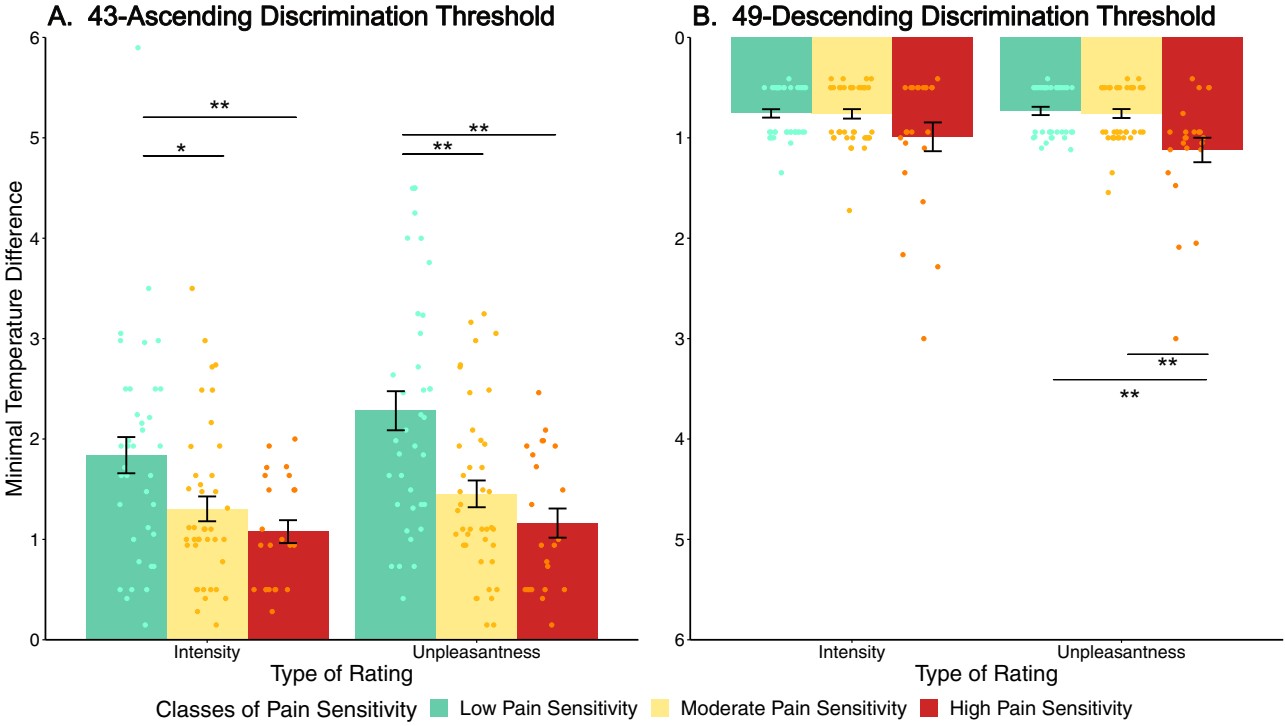

**Fig. 1 Individual differences in the subjective experience of pain are further supported by differences between pain sensitivity classes in their ability to discriminate small differences in stimulus temperature via subjective reports.** Compared to participants from the High (red bars) and Moderate Pain Sensitivity classes (yellow bars), participants from the Low Pain Sensitivity class (green bars) needed a significantly greater increase in temperature from 43 °C to report changes in their perceived pain intensity and unpleasantness ($p$ values: low – moderate: $p_{intensity} = 0.017$, $p_{unpleasantness} = 0.003$; low – high: $p_{intensity} = 0.009$, $p_{unpleasantness} = 0.0005$), as shown by post-hoc two-sided Dunn's tests (**A**). Conversely, the highly sensitive class required a significantly larger decrease in temperature from 49 °C to report a change in perceived unpleasantness in relation to low or moderate sensitivity classes ($p$ values: low – high: $p_{unpleasantness} = 0.001$; moderate – high: $p_{unpleasantness} = 0.003$), as shown by post-hoc two-sided Dunn's tests (**B**). The smallest temperature changes to achieve discrimination in sensation from the sensation of a reference temperature, i.e., 43-ascending discrimination thresholds and 49-descending discrimination thresholds, are represented on the y axis. Error bars represent standard error of the mean. * represents $p$ values < 0.05; ** represents $p$ values < 0.01. Source data are provided as an xlsx Source Data file. $n = 101$ participants.

($r = 0.13$, $p > 0.1$; Supplementary Fig. 4). Consistent with the lack of a relationship, results showed a high cross-validation error between observed and predicted pain ratings: 7.14 VAS units.

The relationship between brain activation in response to high intensity heat stimulus and pain perception was then assessed by comparing brain activation between classes of pain sensitivity. Results of the *F*-tests and the *t*-tests identified no difference between classes in terms of brain activation associated with noxious heat stimulation at a z threshold of 3.1.

**Relationship of reported cold pain intensity with brain activation**. The univariate analysis repeated on our control nociceptive condition, i.e., cold pain, yielded similar results as in the heat paradigm. Significant brain activation was detected in response to high intensity cold stimuli: increased activation was detected in areas including the putamen, the caudate nucleus, the secondary somatosensory cortex, the insula, the anterior cingulate cortex, and the dorsolateral prefrontal cortex, while decreased activation was detected in areas including the amygdala and hippocampus, the primary somatosensory cortex, the posterior cingulate cortex and the precuneus. However, no relationship with individual pain intensity ratings could be identified (Fig. 3, Supplementary Tables 4 and 5). Thus, the findings from heat pain generalized to cold pain.

**Relationship of reported auditory intensity with brain activation**. The main GLM analysis was also performed on our auditory control modality to assess the relationship between brain activation in response to high intensity stimulus and individually perceived intensity in a non-noxious paradigm.

Intensity ratings of high intensity non-noxious auditory stimuli ranged widely across individuals from 0 to 7.94 (average ± SD: 1.01 ± 1.53) (Fig. 4A).

Increased brain activation in response to high intensity stimuli in areas such as putamen, caudate nucleus, primary auditory cortex (AI), SII, insula, and ACC, and decreased brain activation in areas such as amygdala, hippocampus, SI, PCC and precuneus (Fig. 4B, Supplementary Tables 6 and 7).

In sharp contrast with heat pain and cold pain, an effect of individually reported intensity in response to high intensity non-noxious auditory stimulus was identified in areas associated with changes in brain activation in response to high intensity non-noxious auditory stimuli (Fig. 4C, Supplementary Tables 8 and 9). In particular, greater reported auditory intensity was associated with greater activation in AI and the left insula. Conversely, greater reported auditory intensity was associated with greater deactivation in the PCC and precuneus.

**Within-individual effects of heat stimulus intensity**. To ensure that the lack of relationship between brain activation in response to high intensity heat stimulus and individual pain intensity ratings was not due to a lack of sensitivity in the paradigm, within-individual t-tests were performed to investigate differences in brain activation in response to small changes in stimulus intensity, i.e., 1 °C.

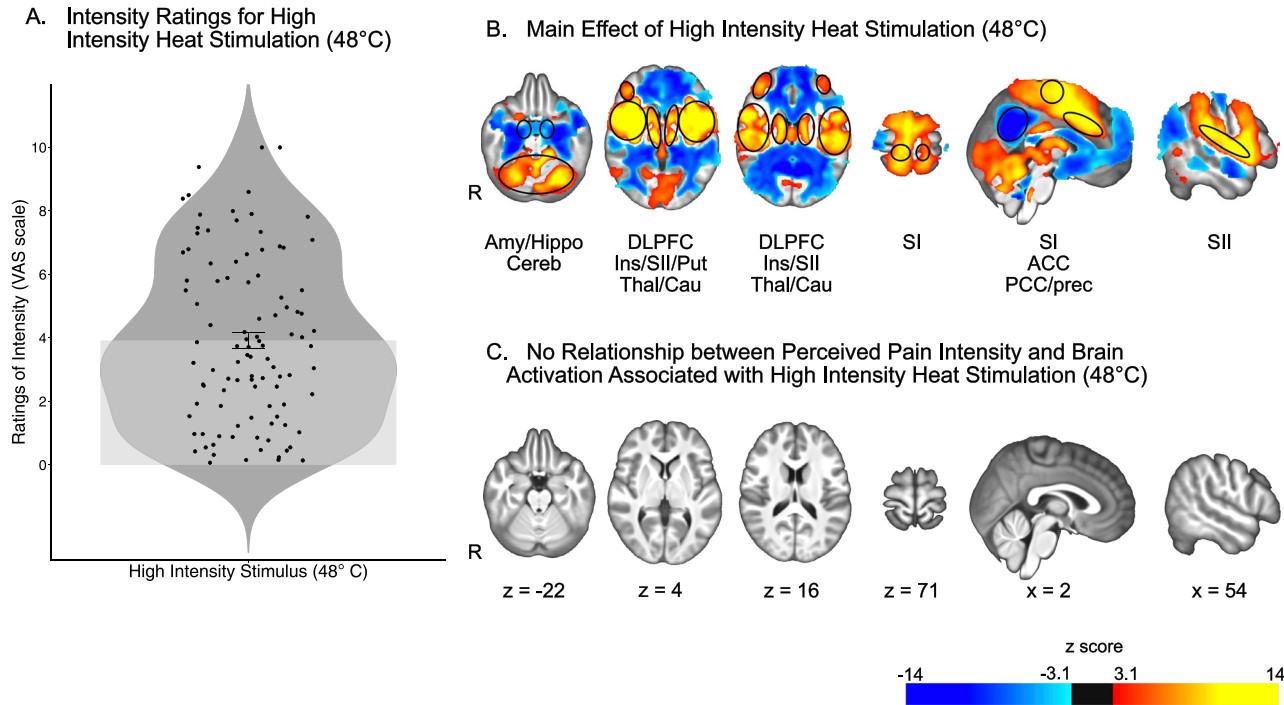

**Fig. 2 Univariate fMRI analysis reveals widespread brain activation associated with high intensity heat stimulation, but no relationship with perceived pain intensity. A** Individual ratings of pain intensity in response to high intensity heat stimulation ranged from 0.07 to 10, with most people providing averaged ratings of pain intensity below 5. Mean and standard error represented by the bar plot. $n = 101$ participants. **B** Effect of high intensity heat stimulation on brain activation. Areas of increased activation included cerebellum (Cereb), thalamus (Thal), putamen (Put), primary somatosensory cortex (SI), secondary somatosensory cortex (SII), insula (Ins), anterior cingulate cortex (ACC), and dorsolateral prefrontal cortex (DLPFC). Areas of decreased activation included the amygdala (Amy) and hippocampus (Hippo), as well as posterior cingulate cortex (PCC) and precuneus (Prec). **C** There was no relationship between perceived pain intensity and brain activation associated with high intensity heat stimulus. Source data are provided as an xlsx Source Data file and activation maps are available on Github.

Ratings of pain intensity reported during high intensity noxious heat stimulus were significantly different from those reported during low intensity noxious heat stimulus: t (100) = 11.719, $p < 0.0001$ (Fig. 5A).

An analysis of changes in brain activation associated with low intensity noxious stimulus revealed changes in similar areas as high intensity noxious stimuli (Fig. 5B), i.e., amygdala, hippocampus, putamen, caudate nucleus, thalamus, SI, SII, insula, ACC, PCC, precuneus, and DLPFC (Fig. 5C, Supplementary Tables 10 and 11). Analyses of the differences in brain activation evoked by high and low intensity heat stimuli (Fig. 5D, Supplementary Tables 12 and 13) confirmed significant differences in the same brain areas, despite the substantial overlap of activation and deactivation.

Analyses performed using the multivariate NPS on our heat task further supported the results of the difference between high and low intensity stimuli by showing a significant within-individual difference in the NPS expression between the two stimulus intensities: t (100) = 6.24, $p < 0.0001$. (Fig. 6).

**Within-individual effects of cold stimulus intensity**. Ratings of reported pain intensity between high and low intensity cold stimuli were not significantly different: t(72) = 1.23, $p = 0.2$ (Supplementary Fig. 5A). Similar to the high intensity noxious cold stimulus (Supplementary Fig. 5B), the low intensity noxious cold stimuli evoked changes in the amygdala, hippocampus, putamen, caudate, insula, SII, SI, ACC, PCC, precuneus, and DLPFC (Supplementary Fig. 5C, Supplementary Tables 14 and 15). Furthermore, significant within-individual differences in activation between the two stimulus intensities occurred in the same areas

(Supplementary Fig. 5D, Supplementary Tables 16 and 17). Thus, in the case of cold pain, the neuroimaging data could detect differences in stimulus intensity that were not detectable in pain ratings.

**Within-individual effects of auditory stimulus intensity**. Similar within-individual t-tests were performed on our non-nociceptive control condition, i.e., auditory stimulus. Ratings of reported intensity between high and low intensity non-noxious auditory stimuli did not differ significantly: t(96) = 1.68, $p = 0.1$ (Supplementary Fig. 6A). Low intensity non-noxious stimuli were associated with changes in similar brain areas as those of the high intensity non-noxious stimuli (Supplementary Fig. 6B), i.e., amygdala, hippocampus, putamen, caudate nucleus, insula, SI, SII, AI, ACC, PCC, precuneus, and DLPFC (Supplementary Fig. 6C, Supplementary Tables 18 and 19). Significant within-individual differences in brain activation in response to the two stimulus intensities were observed in the same areas (Supplementary Fig. 6D, Supplementary Tables 20 and 21).

**White matter deactivation**. Throughout the fMRI results, a significant deactivation could be observed in the white matter, even after preprocessing of the data using FIX. This deactivation suggests that there is a small and constant effect of the stimulation on the white matter and/or global signal intensity (see Supplementary Discussion for more information). lt has been previously shown that inclusion of global signal confounds, which are correlated to the experimental paradigm, i.e., stimulation, might affect the relationship between the paradigm and the resulting brain activations[37]. However, given that the correlation

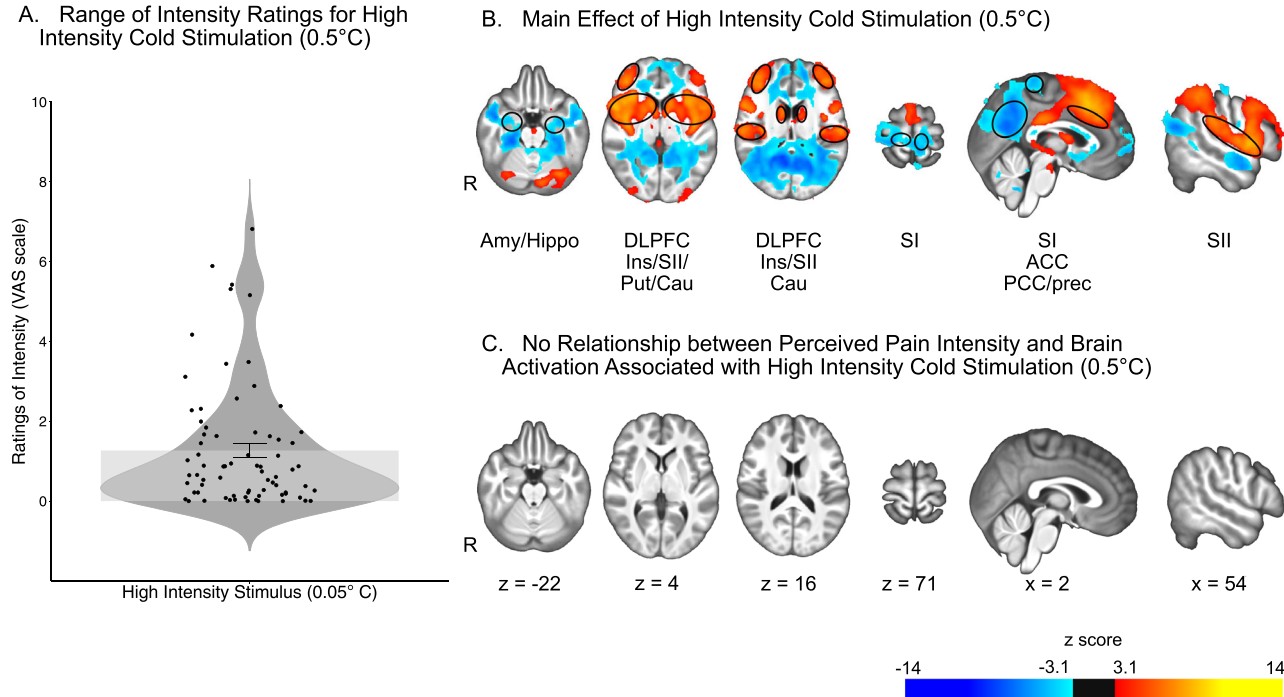

**Fig. 3 Univariate analysis revealed widespread brain activation associated with high intensity cold stimulation (0.5 °C), but no relationship with perceived pain intensity. A** Ratings of pain intensity in response to high intensity cold stimuli ranged from 0 to 6.82. Mean and standard error represented by the bar plot. $n = 73$ participants. **B** Effect of high intensity cold stimulation on brain activation. Areas of increased activation included the putamen (Put), caudate nucleus (Cau), secondary somatosensory cortex (SII), insula (Ins), anterior cingulate cortex (ACC), and dorsolateral prefrontal cortex (DLPFC). Areas of decreased activation included bilateral amygdala and hippocampus (Amy/Hippo), primary somatosensory cortex (SI), posterior cingulate cortex (PCC) and precuneus (Prec). **C** There was no relationship between perceived pain intensity and brain activation associated with high intensity cold stimulation. Source data are provided as an xlsx Source Data file and activation maps are available on Github.

of global signal confounds and our stimulation is close to null across participants, this effect is likely to be small and individualized.

**Effect of head motion**. Two Kruskal–Wallis analyses were performed to test the relationship between head motion in the scanner and pain sensitivity classes. Results from these analyses excluded such a relationship (RMS: $\chi^2$ (2) = 2.3231, $p = 0.3$; FWD: $\chi^2$ (2) = 2.5739, $p = 0.3$), suggesting that head motion was not responsible for the lack of effect of reported pain intensity on brain activation associated with high intensity heat stimuli.

**Effect of repeated stimulation**. An additional Kruskal–Wallis analysis was performed to investigate the stability of pain ratings across multiple stimulus presentations within-individuals during the scanning session. Results of this analysis revealed no effect of time (e.g., repeated stimulation) ($\chi^2$ (1) = 1.7925, $p = 1$) on reported pain intensity.

**Discussion**
In sharp contrast with previous findings[1,3,4,6–8], interindividual differences in subjective reports of pain intensity were not parametrically related to objective assessments of brain activation during noxious heat or noxious cold stimulation in this study. This dissociation is striking given 1) the strong BOLD response associated with noxious stimulation, 2) the detection of within-individual differences in activation evoked by different intensities of noxious stimuli, and 3) the detection of brain activation related to interindividual differences in innocuous auditory perception.

Subjective reports of pain intensity and unpleasantness ranged widely across individuals despite the use of stimuli of fixed intensities. The wide range of pain ratings is in agreement with previously published findings, which have reported substantial individual differences in pain sensations[1,2]. A Gaussian mixture model analysis of pain ratings revealed individuals are distributed across three classes of pain sensitivity, i.e., Low, Moderate, and High Pain Sensitivity.

Multiple converging lines of evidence indicate that the individual differences in subjective reports of pain largely reflect true individual experiential differences in pain magnitude instead of rating biases. First, we found no differences in the psychological and demographic profile between classes. Thus, participants from the Low, Moderate, and High Pain Sensitivity classes did not differ on depression, anxiety, catastrophizing or other variables that may have consistently affected the rating process, resulting in a bias in the reporting and communication of the individual pain experience. These results were further supported by correlation analyses showing no relationship with these variables and pain intensity (supplementary analysis). Second, across all three classes, the ability to report changes in reported pain sensation with a change in temperature as small as 1 °C confirm that individuals could reliably report small differences in stimulus intensity by providing ratings of pain intensity and pain unpleasantness using VAS. Third, classes of pain sensitivity differed in the manner in which they reported their pain sensations evoked by small differences in stimulus intensity. Specifically, participants from the High Pain Sensitivity class reported greater pain intensity and unpleasantness after a smaller increase in temperature from a reference temperature of 43 °C compared to participants from the other classes. Conversely, participants from the High Pain Sensitivity class required a greater decrease in temperature from a reference temperature of 49 °C to report lower pain unpleasantness, but not intensity, ratings than participants from the other

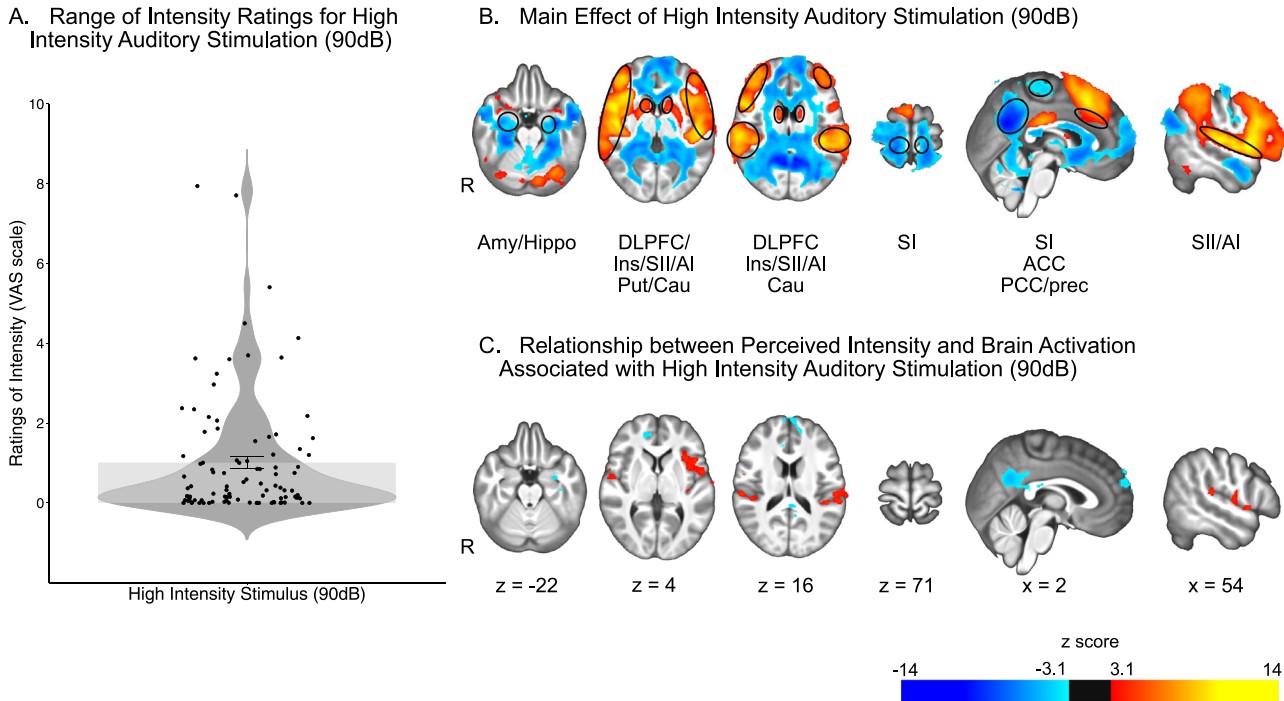

**Fig. 4 Univariate analysis revealed widespread brain activation associated with high intensity auditory stimulation (90 dB) and related to perceived intensity. A** Ratings of perceived auditory intensity in response to high intensity auditory stimuli ranged from 0 to 7.94. Mean and standard error represented by the bar plot. $n = 97$ participants. **B** Effect of high intensity auditory stimulation (90 dB) on brain activation. Areas of increased activation included putamen (Put), caudate nucleus (Cau), secondary somatosensory cortex (SII), primary auditory cortex (AI), insula (Ins), anterior cingulate cortex (ACC), and dorsolateral prefrontal cortex (DLPFC). Areas of decreased activation included bilateral amygdala and hippocampus (Amy/Hippo), primary somatosensory cortex (SI), posterior cingulate cortex (PCC) and precuneus (Prec). **C** High perceived intensity was associated with a greater increase in brain activation associated with high intensity stimulation in areas such as AI, SII, and insula and a greater decrease in brain activation in PCC and precuneus. Source data are provided as an xlsx Source Data file and activation maps are available on Github.

classes. This lack of statistically significant results in the pain intensity ratings could be driven by the smaller difference in temperature decrease needed to achieve perceptual changes in pain intensity compared to pain unpleasantness between the Low and High Pain Sensitivity classes. These results are congruent with correlations between discrimination thresholds and pain ratings (Supplementary analysis). Taken together, the psychophysical results support VAS ratings of pain intensity and pain unpleasantness as a reliable measure of the subjective experience of pain.

High intensity noxious heat stimuli were used to assess positive and negative changes in brain activation associated with reported pain intensity. Consistent with a large body of prior evidence[1,9,14,17,38–45], noxious heat stimulation evoked robust increases and decreases in brain activation in areas that were previously described as being associated with nociceptive processing and pain. Despite this effect, no relationship between reported pain intensity and brain activation or deactivation was detected. This is even more surprising given the wide range of reported pain intensity ratings from 0.07 to 10.

The results of the massive univariate analysis were further confirmed by assessment of activation using the NPS and a LASSO-PCR model. All 101 participants exhibited expression of the NPS during high intensity noxious heat stimulation. However, no relationship was detected between the expression of the NPS and ratings of pain intensity. The LASSO-PCR model showed no relationship between individual pain intensity ratings and brain activation associated with high intensity heat stimuli. Thus, even with approaches more comprehensive than the massive univariate analyses, brain activation supporting individual differences in reported pain was not identified.

Because of the discrepancy between our current results and prior findings[1,3,4], we performed several complementary analyses to eliminate potential explanations for the lack of relationship between reported pain intensity and changes in brain activation. First, there was no relationship between head motion and pain sensitivity classes. As such, differential signal loss induced by movement cannot account for the lack of relationship between brain activation and individual differences in pain ratings. Second, analyses of effects of repeated stimulation revealed that sensitization and/or habituation did not differ between classes. Thus, differential responses across repeated stimulation could not account for the lack of relationship between brain activation and individual differences in pain ratings. Third, the massive univariate analysis of the heat paradigm was repeated in supplementary analyses with the following conditions: (1) within classes of Pain Sensitivity, (2) while including all the psychological and demographic recorded data and head motion parameters, (3) while including only the adult participants of the cohort, and (4) while selecting a subsample of participants matching the one from the 2003 study[1]. These supplementary analyses further confirmed the lack of a relationship between brain activation associated with noxious heat stimuli and individual perceived pain intensity. Fourth, a supplementary Bland-Altman analysis revealed a small but systematic decrease (1.5 VAS units) in pain intensity ratings from the first to the second session. This decrease could be due to an exposure effect or to the MRI environment during the second session. Nevertheless, given that our main univariate analysis was performed on brain imaging data and rating data acquired during the same session (MRI session), this decrease in ratings between sessions would not impact our findings.

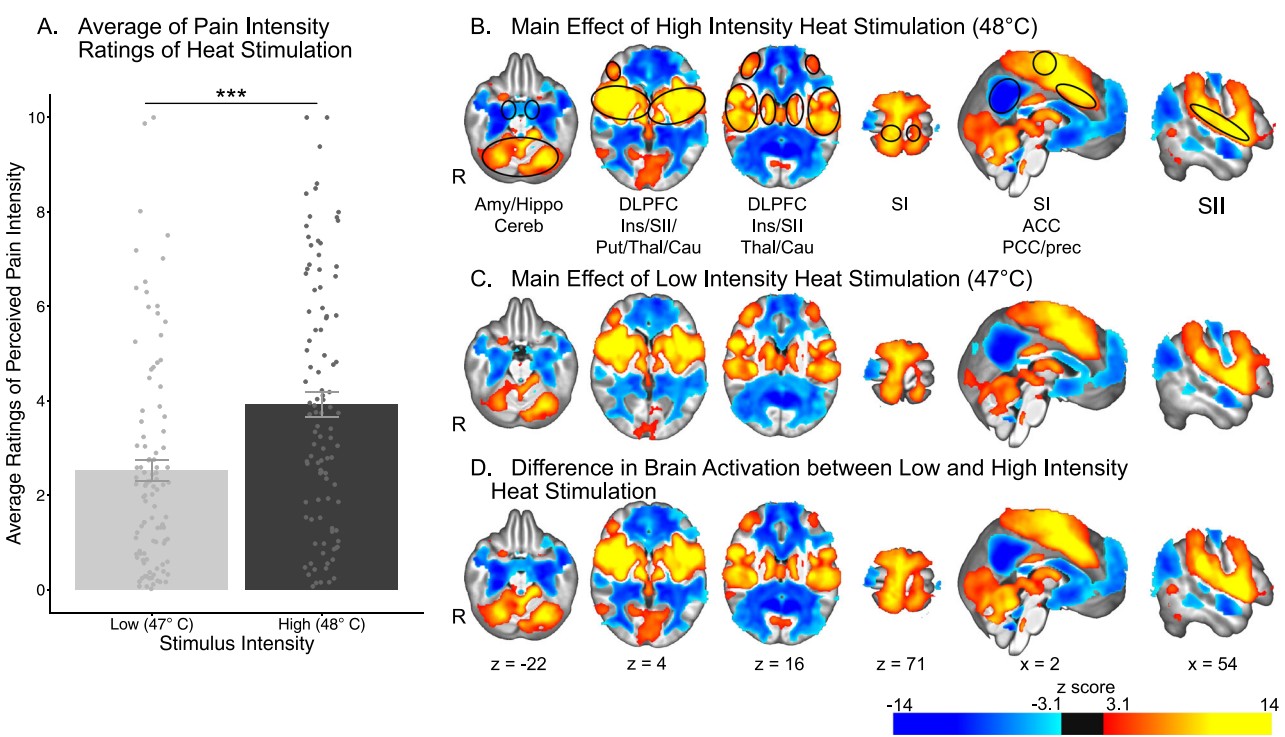

**Fig. 5 Effect of graded increases in intensity of heat stimulation on brain activation. A** Average ratings of pain intensity associated with heat stimulation significantly differed between high and low intensity stimulation, as shown by a two-sided paired sample t-test ($p = 2.2\ e^{-16}$). $n = 101$ participants. Increased brain activation in response to high (**B**) and low (**C**) intensity heat stimulation and differences between the two intensities of stimuli (**D**) are observed in areas such as the putamen (Put), the caudate nucleus (Cau), the thalamus (Thal), the primary somatosensory cortex (SI), the secondary somatosensory cortex (SII), the insula (Ins), dorsolateral prefrontal cortex (DLPFC), and the anterior cingulate cortex (ACC). Decreased activation in response to the same stimuli is especially present in the precuneous (Prec) and the posterior cingulate cortex (PCC). Error bars represent standard error of the mean. *** represents $p$ values < 0.001. Source data are provided as an xlsx Source Data file and activation maps are available on Github.

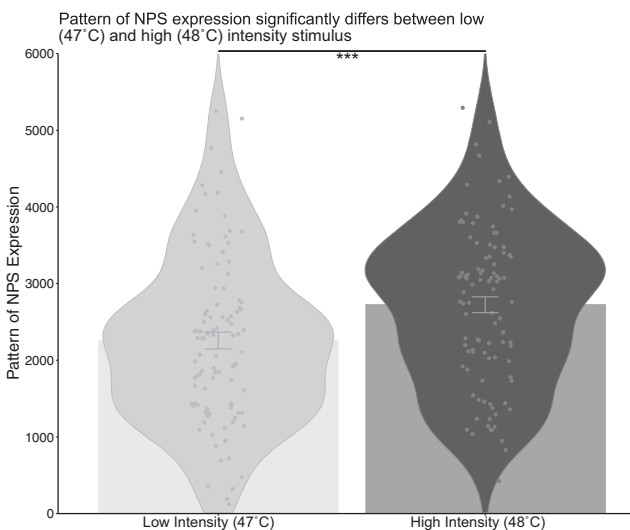

**Fig. 6 NPS expression significantly differs between low (47 °C) and high (48 °C) intensity heat stimuli, as shown by a two-sided paired sample t-test ($p = 1.021\ e^{-08}$).** The average NPS expression, represented by the bars, is higher in high intensity heat stimulation compared to low intensity heat stimulation. In addition, the violin plots, which represent the distribution of individual NPS expression in each intensity, show a greater density of individuals with higher NPS expression in the high intensity heat stimulation than in the low intensity heat stimulation. Finally, the error bars represent one standard error of the mean. *** represents $p$ values < 0.001. $n = 101$ participants. Source data are provided as an xlsx Source Data file.

Furthermore, differences in the experimental paradigms could be the source of the differences between the current results and those published in 2003[1]. First, the noxious heat stimuli used in the present investigation were shorter (10 s plateau) than the longer duration stimuli (30 s plateau) used previously[1]. However, despite the different durations, robust brain activation during nociceptive processing was highly evident in the univariate analysis and was detected by expression of the NPS within all 101 participants. Furthermore, the mean and range of pain intensity ratings between the two studies, i.e., the current study and the one reported in Coghill et al.[1], were similar (current study: mean = 3.92, range = 0.07–10.0; 2003 study: mean = 4.79, range = 1.05–8.9) despite the different durations of stimulation. Thus, small parametric differences of the stimuli, such as its duration or intensity, could have had a differential impact on the sensitivity of the BOLD effect to detect individual differences in pain versus nociceptive processing. Additionally, experimental differences, including using different scanners and different acquisition sequences, might also underlie the dissociation in the findings. Regardless, if small differences in experimental parameters can dramatically affect resulting brain activation while evoking similar pain ratings in response to these stimuli, any definition of brain markers based on such results would be, at best, unreliable. Finally, the early study[1] had a very small number of participants (6 highly sensitive and 6 low sensitive participants) vs. the greater number (101) of participants in the present investigation (23 highly sensitive, 41 moderately sensitive, and 37 low sensitive participants). As such the early findings could be susceptible to errors arising from small group effects[32].

The absence of a relationship between individual differences in reported pain intensity and brain activation was also replicated

for pain evoked by noxious cold. During cold stimulation there was again a robust activation of numerous brain areas associated with nociceptive processing. However, no regions exhibited a relationship with individual reports of pain intensity. Similar to our results in the heat paradigm, ratings of reported cold pain intensity covered a wide range of the VAS, rendering unlikely that the lack of effect would be due to too little variability in the ratings.

In contrast with both the noxious heat and noxious cold data, analysis of the auditory data reveals that the regression approach was sufficiently sensitive to detect a relationship between individual differences in reported auditory intensity. Presentation of the 900 Hz sawtooth waveform tone produced robust activation of areas important in auditory processing including AI, the insula, the ACC, and the PCC. Of these, AI and the insula exhibited a positive relationship with individual differences in reported auditory intensity while the PCC and the precuneus exhibited a negative relationship with reported auditory intensity. The range of ratings of individually reported pain intensity in response to the high intensity non-noxious auditory stimuli was similar to the one in response to the high intensity noxious cold and heat stimuli. This further suggests that the lack of results in the heat and cold series is not due to an insufficient variability in the individually reported pain intensity. Moreover, the auditory data confirm that the fMRI paradigms were sufficient to detect individual differences in reported intensity. This suggests that the lack of relationship between brain activation and reported intensity is inherent to a pain-specific mechanism that remains to be defined.

To further confirm the sensitivity of our fMRI paradigms, we analyzed brain activation associated with within-individual differences in stimulus intensity. All three modalities of stimulation produced robust changes when brain activation during high intensity stimulation was compared to that of low intensity stimulation. These results confirm that our paradigm was sufficient to detect changes in brain activation elicited by small differences in stimulus intensity. In addition, these within-individual differences in the cold and auditory paradigms were detected despite a lack of statistically significant difference in reported pain intensity between low and high intensity stimuli.

Interestingly, this disconnect between the cerebral representations of within- and interindividual pain perception has been previously observed and reported using EEG[46]. In that EEG study, within-individual differences in nociception were associated with changes in EEG responses, while interindividual differences in pain were not associated with changes in EEG responses. These previous findings further support our present findings. These results taken together suggest that brain activation might be more associated with nociception than with individual differences in perceived pain.

In conclusion, the present findings provide further evidence that different individuals can instantiate markedly different experiences of pain from the same sensory inputs. Furthermore, these results strongly suggest that interindividual differences in pain ratings reflect true experiential differences rather than reporting biases. The dissociation between the subjective reports of pain and underlying brain activation raises significant questions about the ability of BOLD activation-based fMRI to serve as a biomarker for pain intensity. However, recent findings suggest that BOLD connectivity-based measures could provide such biomarkers in chronic pain[47]. This dichotomy between activation- and connectivity-based measures of pain raises significant questions about the ability of BOLD fMRI to adequately capture neural activity involved in the instantiation of the pain experience, despite its efficacy in capturing central processes of nociception. Are brain mechanisms supporting the construction of an experience of pain so globally dispersed and divorced from

nociceptive mechanisms that their patterns of activity represent such a radical transformation of afferent input that there is no longer a monotonic relationship between perceived pain and the firing of central nervous system neurons supporting that experience? This notion starkly contrasts with the idea that the neural signals that instantiate pain are sufficiently degenerate/redundant such that the experience can remain in the face of substantial damage to nociceptive processing regions in the brain[9,17,48]. Perhaps the resilience suggested by lesion studies[49,50] may reflect the ability of pain to be instantiated by multiple distinct systems, and this multiple realizability may be more important across healthy individuals than previously thought.

In summary, while fMRI may have significant potential to provide insight into mechanisms of acute and chronic pain and potential trajectory of treatment responsivity[51–55], the long-standing clinical dictum, "pain is what the patient says it is," remains. Our results suggest that the use of fMRI as an objective measure to infer reported pain intensity for medico-legal purposes may need to be considered with great caution.

## Methods

**Participants.** 143 healthy individuals (58 males and 85 females, age: $28 \pm 7.2$, mean ± SD, age range: 14–44 years old) in an ongoing study underwent Quantitative Sensory Testing (QST) and neuroimaging and completed behavioral and psychological surveys. Data from 34 participants (12 males and 22 females, age: $28 \pm 6.1$, mean ± SD), which were not used in prior analyses or in the analyses described here, were used to train an automated classifier (FSL FIX, FMRIB's ICA-based Xnoiseifier, FSL, Oxford, UK)[56,57] that was used to denoise the fMRI data. 4 of these 34 participants had subtle incidental findings reported by a radiologist, that due to their nature and localization should not affect the training of the FIX classifier. Out of the remaining 109 participants, 8 participants were excluded because of insufficient quality of the fMRI images or incidental findings of abnormalities on MRI. The remaining 101 healthy volunteers (43 males and 58 females, age: $28.5 \pm 7.7$, mean ± SD) were included in the analyses of the heat series described below. Due to technical issues during the MRI session, 28 of these participants had missing data in the cold series and 4 had missing data in the auditory series.

Participants and parents/legal guardians of minor participants gave their written informed consent and minors provided written assent in accordance with the institutional review board of Cincinnati Children's Hospital Medical Center, which approved the study. Participants received monetary compensation for their participation in the study. Exclusion criteria included active neurological or psychiatric disorder that impacted the participant's ability to perform the tasks requested, the presence or history of chronic pain, medications that could interfere with QST or brain function, positive screen for recreational drugs, any serious pathology, substantial uncorrected visual deficit, and any MRI contraindication, such as any metallic implant or braces.

**General design.** Participants in this study completed two sessions: a QST and an MRI session. Sessions were on average 36 days (SD: 50.3) apart. During the QST session, participants first were familiarized with pain rating tasks by evaluating a set of heat stimuli before experiencing the heat, cold, and auditory stimuli included in the MRI session. In addition, participants completed psychological surveys. This session lasted approximately three hours. During the MRI session, participants received and rated heat, cold, and auditory stimuli. The MRI session lasted approximately 90 min.

All thermal stimuli were generated and controlled by a Pathway model ATS (Medoc, Ramat Yishai, Israel). A $16 \times 16$ mm thermode was used for the heat stimuli and a $30 \times 30$ mm thermode for the cold stimuli. During all the thermal stimuli, participants were instructed to keep their skin in contact with the thermode as long as they could tolerate the temperature.

The auditory stimuli were played using iTunes (Apple Inc, Cupertino, CA, USA). During the QST session, participants wore over-the-ear Puro Soundslab BT5200 calibrated headphones. During the MRI session, participants listened to the sounds through MRI-compatible headphones.

For both sessions, minor participants were accompanied by their legal guardians. During the QST session, legal guardians were asked to be present during the consenting part and to confirm the participant's eligibility. During the MRI session, legal guardians were asked to be present to confirm MRI compatibility. Legal guardians then stepped out of the room during the testing protocols to avoid parental influence on the participant's responses[58–60].

All participants were asked to turn their cell phone off to avoid any distraction during the testing protocol. Cell phones were kept secured, away from the subjects along with other ferromagnetic objects during the MRI session.

**Visual analog scales**. Throughout the two sessions of this study, participants reported pain intensity and pain unpleasantness on visual analog scales (VAS). The VAS for pain intensity was anchored with the words 'no pain sensation' and 'most intense pain sensation imaginable', while the VAS for pain unpleasantness was anchored with the words 'not at all unpleasant' to 'most unpleasant imaginable'[34,35]. The anchors of the VAS to rate auditory intensity were 'not at all loud' and 'the loudest imaginable', while VAS to rate auditory unpleasantness ranged from 'not at all unpleasant' to 'the most unpleasant imaginable'. While participants were familiarized with the heat stimuli in the QST session, they rated their sensations by positioning a slider in a plastic scale until it matched the level of their sensation. Numbers on the back of the scale allowed the experimenter to read the level of their sensations (range 0 to10). When participants were trained on the tasks they would experience in the scanner and during the MRI session, they reported their sensations using two computerized VAS scales, which were controlled with the IDL software (L3Harris Geospatial, Broomfield, CO, USA). In the computerized version of these scales, participants slid the cursor by using a trackball mouse (QST session: Logitech, Newark, CA, USA; MRI session: Current Designs, Philadelphia, PA, USA).

VAS scales for the measurement of pain have been previously shown to be true ratio scales[34]. In addition, the VAS is sensitive to changes in noxious temperature as small as 0.2 °C[61]. The ability of participants to distinguish and rate pain intensity and unpleasantness on two different VAS scales has been previously confirmed: Price et al.[34] showed that ratings of pain intensity and unpleasantness resulted in distinct stimulus-response curves. Furthermore, using extreme anchors, such as 'most intense pain imaginable' or 'most unpleasant pain imaginable', ensures the reliability of the scale as a ratio scale[34].

These scales have been previously used in studies investigating individual differences in the experience of pain and shown to capture such differences[1,19,62,63]. Finally, of importance in this study including participants under the age of 18, VAS scales have been validated as reliable measures of pain in children[64].

### QST session

*Familiarization*. During the first session, participants were familiarized with heat stimuli by receiving 32 5-second heat stimuli on their left forearm. For each stimulus, the temperature increased from a baseline of 35 °C at a rate of 6 °C/s, plateaued for 5 s and returned to baseline at a rate of 6 °C/s. These stimuli included four repetitions of the following eight temperatures in a pseudo-randomized order: 35 °C, 43 °C, 44 °C, 45 °C, 46 °C, 47 °C, 48 °C, and 49 °C. All participants received the stimuli in the same order. After each stimulus, participants were instructed to rate their reported pain intensity and unpleasantness on the two VAS scales described above. The inclusion of a familiarization session during which participants experience and rate multiple short-duration heat stimuli has been shown to minimize the session-to-session order effect, hence increasing the reproducibility of the ratings[65]. Given that this study aims at investigating brain mechanisms underlying individual differences in the experience of pain assessed by psychophysical measurements and that psychophysical measurements and fMRI data were acquired in separate sessions, ensuring the reproducibility of the ratings was essential to confirm that we always have a large range of pain sensitivity and can address stability across sessions.

*fMRI training stimuli*. Participants were then trained on the heat, cold, and auditory tasks they would perform during the MRI session. Each task included 10-second long stimuli of two different intensities, followed by a 16-s rating period and a 22-s resting period. During the heat and cold tasks, stimuli were delivered to the back of the lower left leg. The heat task included six high intensity noxious stimuli (48 °C), and one low intensity noxious stimulus (47 °C). Heat stimuli had the same increase and decrease rates as the ones in the familiarization phase of this session.

The cold task included 4 high intensity noxious stimuli (0.5 °C) and one low intensity noxious stimulus (3 °C). During the cold stimuli, temperature decreased from a baseline of 35 °C at a rate of 3 °C/s, plateaued for 10 s, and returned to baseline at a rate of 6 °C/s.

The auditory task included 5 high intensity non-noxious stimuli (90 dB) and 2 low intensity non-noxious stimulus (80 dB). The auditory stimuli were designed as 900 Hz sawtooth waves with an overall time course similar to the heat stimuli. In the high intensity stimuli, the sound increased from silence to 90 dB in 2.2 seconds and plateaued for 10 s before returning to 0 dB at the same rate. In the low intensity stimuli, the sound increased to 80 dB in 2 seconds and plateaued for 10 seconds before returning to 0 dB at the same rate. These stimulus parameters were chosen to provide a non-somatic innocuous control condition.

During the heat and cold tasks, participants were instructed to rate their reported pain intensity and unpleasantness after each thermal stimulus. In-between the presentation of the scales, a black screen was displayed. For the auditory stimuli, participants were instructed to rate the intensity and unpleasantness of the sounds that they experienced.

*Behavioral/psychological measures*. All behavioral/psychological measures were recorded directly in the web-based software REDcap.

*All participants*. Finally, during this session, participants or their legal guardian provided their medical history and demographics information, including race and education level.

All participants then completed the Edinburgh Inventory to define handedness[66,67]. The modified version of the survey used in this study included 12 items that were used to calculate a laterality quotient ranging from −100 to +100. Positive quotient indicated a right-hand dominance, while negative quotient indicated left handedness.

In addition, participants completed surveys, which assess psychological factors known to impact pain sensations. In all participants, sleep patterns were assessed using the Epworth Sleepiness Scale[68] and the Pittsburgh Sleep Quality Index[69]. The Epworth Sleepiness Scale is an 8-item questionnaire measuring daytime sleepiness. Participants report their likeliness of dozing during activities for each item on a 4-point Likert-type scale, ranging from "0 = would never doze" to "3 = high chance of dozing". Responses to these items are added, resulting in a score ranging from 0 to 24 with lower score indicating less doziness. The Pittsburgh Sleep Quality Index was modified to include 18 self-rated items and results in a score ranging from 0 to 21 with higher scores signifying greater sleep difficulties.

All participants completed the 14-item Freiburg Mindfulness Scale[70], by rating their mindfulness experienced on a 4-point Likert-type scale ranging from "rarely" to "almost always". Scores on this scale range from 1 to 56.

The emotional state of all participants was assessed through the completion of the Barratt Impulsiveness Scale[71] and the Positive and Negative Affect Schedule (PANAS)[72]. The Barratt Impulsiveness Scale includes 30 items rated on a 4-point Likert-type scale ranging from "rarely/never" to "almost always/always" and resulting in a score between 30 and 120 with higher score indicating greater impulsiveness. The PANAS includes 20 items describing positive and negative emotional states and rated on a 5-point Likert-type scales ranging from "Very slightly or not at all" to "Extremely". Items are then split into positive and negative affect and used to calculate a positive affect and a negative affect score. Each score can range from 10 to 50 with higher score indicating greater level of affects.

Functional Disability were evaluated using the Functional Disability Inventory[73]. Participants evaluate the 15 items of this scale on a 5-point Likert-type scale, ranging from "no trouble" to "impossible". Participants' responses are used to calculate a total score ranging from 0 to 60 with higher scores indicating greater functional disability.

Finally, all participants completed a modified version of the Experience of Discrimination survey[74], which included 26 items and results in a score ranging from 0 to 72.

*Adult participants*. Emotional state of adult participants was assessed with the PROMIS anxiety, depression, and pain interference scales[75–77].

Pain catastrophizing was assessed using the Pain Catastrophizing Scale[78]. This scale includes 13 items that are evaluated on a 5-point Likert-type scale ranging from "not at all" to "all of the time". Responses to these scales are used to compute a score ranging from 0 to 52, with greater score indicating greater catastrophizing.

*Adolescent participants*. Minor participants completed the pediatric versions of the three PROMIS scales[79–81], the Screen for Child Anxiety-Related Disorder (SCARED)[82,83] to evaluate their emotional state. PROMIS scales include 8 items each describing situations that participants rate on a 5-point Likert-type scale ranging from "Never" to "Almost always". Scores to these scales range from 8 to 40 with higher score being associated with more emotional disturbances. SCARED includes 41 items. Participants evaluate how true each item is on a scale ranging from "not true or hardly ever true" to "true or often true". A total score ranging from 0 to 42 can be calculated, with scores greater than 25 indicating potential clinical anxiety disorders. In addition, sub-scores for panic disorder, generalized anxiety disorder, separation anxiety disorder, social anxiety disorder, and significant school avoidance can be calculated. Pain catastrophizing was assessed using the Child version of the Pain Catastrophizing scale[84]. Similarly to the adult scale, this scales include 13 items that are evaluated on a 5-point Likert-type scale ranging from "not at all" to "extremely". Responses to these scales are used to compute a score ranging from 0 to 52, with greater score indicating greater catastrophizing.

Surveys assessing psychological factors were completed again during the MRI session.

### MRI session

*MRI acquisition*. During the MRI session, participants laid in a supine position in a Philips 3T Ingenia scanner with a 32-channel head coil. During this session, all participants always underwent a T1 structural scan first. They then completed three BOLD fMRI series of heat stimuli, one fMRI series of cold stimuli and one fMRI series of auditory stimuli. Resting BOLD and arterial spin label (ASL) series were also acquired but are not reported here. The order of the BOLD and ASL series was counterbalanced between participants.

A radiologist inspected the structural images of the participants for incidental findings; there were no major findings in the participants included in the current study.

*T1 structural scan*. The multi-echo (4 echoes) T1-weighted series was acquired using the following parameters: repetition time (TR): 10 ms; echo times (TE): 1.8, 3.8, 5.8, 7.8; flip angle: 8; FOV: 256 × 224 × 200 mm; voxel size: 1 × 1 × 1 mm; slice orientation: sagittal. The total duration of this scan was 4 min 42 s.

*BOLD fMRI*. Each functional image series consisted of 193 volumes acquired using the following parameters: TR: 2 s; TE: 35msec; voxel size: 3 × 3 × 4 mm; FOV: 240 × 240 × 136 mm; slice orientation: transverse; slice order: ascending; dummy scans: 2. Each series lasted 6 min 26 s, after an 8-second pre-scan time.

*Block-design fMRI series*. Participants completed three fMRI series of heat stimuli, receiving a total of 17 high intensity noxious stimuli (48 °C) and 4 low intensity noxious stimuli (47 °C). Two series included 6 high intensity and 1 low intensity noxious stimuli each and one series included 5 high intensity and 2 low intensity noxious stimuli. To limit sensitization or habituation to the stimuli, the position of the thermode was slightly moved on the participant's calf between heat series. The repetition of the heat fMRI series, which was the main task of this study, was meant to increase statistical power at the individual level as well as decrease false positive rates. In addition, participants completed a cold and an auditory series in order to assess the relationship of individual differences in perceived intensity and brain activities in another noxious modality (cold) as well as a non-somatosensory modality (auditory). All the series were derived from the same paradigms as those participants completed during the training part of the QST session. Hence the cold series included 4 high intensity and 1 low intensity stimuli; the auditory series included 5 high intensity and 2 low intensity stimuli. The order of the fMRI series was counterbalanced between participants.

After each stimulus, participants were instructed to rate the intensity and unpleasantness of their sensation on the same computerized VAS scales as in the QST session using an MRI-compatible trackball (Current Designs, Philadelphia, PA, USA).

### Statistical analyses
*Psychophysical analyses*
Classes of pain sensitivity based on reported heat pain sensation: Statistical analyses were performed on pain intensity and unpleasantness ratings of the 5-second stimuli acquired during the familiarization part of the QST session to investigate individual differences in pain sensitivity. Using these data allowed us to investigate individual differences in pain at a psychophysical level and independently from the data used for the fMRI analysis. A mixture model analysis was performed with the stimulus-response data, including pain intensity and pain unpleasantness ratings, in order to objectively group individuals according to pain sensitivity (Mplus 8.4, Los Angeles, CA, USA). Models comprised of 1–5 classes were tested in order to identify the best fit of the grouping model. The selection of the best model was based on two criteria. First, any model that created spurious classes was eliminated. Mixture models including spurious classes were defined as models, in which one of the classes contained 10% or less of the participants included in our dataset. In this case, the mixture solution was considered to be unlikely to reflect real sub-populations or classes of Pain Sensitivity and more likely to reflect an over-extraction of classes[85]. Second, from the remaining models, the one with the lowest Bayesian Information Criterion (BIC) was selected[86,87].

Experimental pain has been previously shown to exhibit a positively accelerating stimulus response relationship that can be fitted with a power function[35]. Exponents and proportionality constant of the power function representing the stimulus-response curve for each class were obtained in three steps: 1) the averaged ratings of intensity and unpleasantness and the difference in temperature from the 35 °C baseline were converted into log values for each individual; 2) a linear regression was performed in double log space; 3) proportionality constants and exponents were extracted from the linear regression model. Given that 35 °C was used as baseline, ratings of this temperature were not used in the calculation of the linear regression model.

To further characterize the classes resulting from the mixture model analysis, univariate chi-squares tests were performed in SPSS (version 25, Armonk, NY, USA), using demographics measures, including sex, race, and economic status, as well as handedness. In addition, Wald Z statistics were used to test age and all behavioral and psychological variables for differences between classes, including scores to the Epworth Sleepiness Scale, Pittsburgh Sleep Quality Index, PROMIS anxiety, depression, and pain interference, Freiburg Mindfulness Scale, PANAS, Pain Catastrophizing Scale, Barratt Impulsiveness Scale, and Experience of Discrimination. Bonferroni correction for multiple tests was applied when appropriate.

Rating-based discrimination thresholds: To confirm that individual differences in pain ratings reflected true experiential differences rather than rating biases, defined as a tendency of an individual to rate a certain way, we sought to determine if discrimination of pain intensity and unpleasantness differed according to pain sensitivity classes. Discriminability of pain was assessed by defining the smallest detected and self-reported increase or decrease in pain from a temperature of reference. As different classes of pain sensitivity were expected to behave differently across the range of temperatures, these analyses were performed at both the low end (43 °C) and the high end (49°) of the noxious range. 43 °C-ascending discrimination thresholds were defined as the lowest temperature > 43 °C that was rated as more

intense or unpleasant than the sensation at 43 °C 50% of the time. Similarly, 49 °C-descending discrimination thresholds were defined as the highest temperature < 49 °C that was rated as less intense or unpleasant than the sensation at 49 °C in 50% of the trials. The following steps were completed to define these thresholds: 1) individual ratings of the stimuli were binarized based on whether they were higher, respectively lower, than the reference value, i.e., rating of 43 °C stimuli for 43 °C-ascending discrimination threshold, respectively rating of 49 °C stimuli for 49 °C-descending discrimination threshold; 2) individual logistic regression was modeled, defining the individual proportionality constants and exponents; 3) rating-based discrimination thresholds were calculated for a probability of 0.5 by dividing -the constant by the exponent. If participants reported differences in their pain intensity or unpleasantness compared to ratings of 43 °C or 49 °C in all the other trials, rating-based discrimination thresholds were defined at an arbitrary value that was closer to the reference than the minimal temperature change. This was to reflect that these participants were likely able to detect a change in temperature smaller than 1 °C. Hence, 43 °C-ascending discrimination thresholds were defined as 43.5 °C and 49 °C-descending discrimination thresholds were defined as 48.5 °C.

43 °C-ascending and 49 °C-descending discrimination thresholds were then compared between classes. Due to violations of the assumption of normality, Kruskal–Wallis tests were performed to investigate differences in 43 °C-ascending and 49 °C-descending individual discrimination thresholds between the classes of pain sensitivity. Post-hoc Dunn tests were performed when appropriate.

These analyses were performed using the software Rstudio version 3.6.2 (Boston, MA, USA).

A significance p threshold was defined at 0.05 in all the psychophysical analyses.

**fMRI analyses**. All MRI data were first inspected for motion and scanner artifacts. They were then preprocessed and analyzed with FSL (FMRIB Software Library, version 6.0.1 Oxford, UK).

**Preprocessing of the MRI data**. Structural images were first corrected for bias using FMRIB's Automated Segmentation Tool (FAST)[88]. Images were then brain extracted using the Brain Extraction Tool (BET)[89] and normalized into standard space MNI-152 using FMRIB's Linear Image Registration Tool (FLIRT)[90,91]. Finally, images were segmented into the different tissue types and white matter and cerebrospinal fluid were masked at a probability threshold of 0.95.

All fMRI data were first registered to the structural scan and then to the standard space MNI-152 using FLIRT and FNIRT (FMRIB's Non-linear Image Registration Tool)[92–95]. Images were then preprocessed in the following steps: motion correction using MCFLIRT (Motion Correction FMRIB's Linear Registration Tool)[91], slice timing correction, brain extraction (BET), spatial smoothing (FWHM: 5 mm) with SUSAN[96], and high pass filter (cutoff: 100 s). Following this, data were split into 25 components by performing a Probabilistic Independent Component Analysis (PICA) using MELODIC (Multivariate Exploratory Linear Optimized Decomposition into Independent Components)[97]. Components resulting from MELODIC were then automatically classified and components identified as noise were removed using a FIX (FMRIB's ICA-based Xnoiseifier, FSL, Oxford, UK)[56,57] classifier trained on fMRI heat series of 34 independent participants. Before running the FIX classifier on the cold and auditory fMRI data, the accuracy of the trained classifier was tested on a subset of these data using FIX. Finally, cleaned filtered images were corrected by intensity normalization.

Data were visually inspected after each preprocessing step to confirm its success.

**Main effect of high intensity stimulus and effect of reported pain intensity**
*Univariate fMRI analyses*. First-level General Linear Model (GLM) analyses were performed on each individual fMRI heat series using FEAT[98]. One block-design regressor of interest, i.e., high intensity stimuli, and two block-design regressors of no interest, i.e., low intensity stimuli and rating periods, were defined. Individual second-level GLM analyses were performed on contrast parameter estimates (COPE) images derived from the previous level using FEAT[99], allowing combination of the individual heat series using a fixed-effect statistical model. Finally, a group-level GLM was performed with FEAT[99] using COPE images from the individual second-level analyses using mixed-effect FLAME 1 and 2 statistical model. In addition, pain intensity ratings of high intensity stimuli were averaged for each individual and were defined as a covariate of interest. These ratings were then mean centered and used to characterize the relationship between reported pain intensity and brain activation evoked by the high intensity stimuli. To ensure that any potential effect of individual pain sensitivity was not due to a between-session difference, only ratings collected during the MRI sessions were used.

To further our understanding of brain activations associated with individual differences we replicated these analyses on the auditory and cold fMRI paradigms. Since these paradigms included one series, only individual first-level and group-level GLM analyses were performed.

*Application of the multivariate Neurologic Pain Signature (NPS)*. To ensure that the lack of relationship between ratings of pain intensity and brain activation associated with high intensity heat stimuli was not due to a lack of sensitivity inherent to the massive univariate nature of the previous analyses, a new analysis using the

NPS[21–23,100–102] was performed. This signature was developed using advanced machine-learning techniques to define a multivariate pattern of brain activation associated with heat pain, which includes the thalamus, the posterior and anterior insula, the secondary somatosensory cortex, the anterior cingulate cortex, and the periaqueductal gray matter, among other regions.

For each participant, a scalar value representing their expression of the NPS in response to heat stimulation was computed. Only voxels included in the original NPS definition were included in this computation. This computation was performed for the high intensity heat stimuli (48 °C) and for the low intensity heat stimuli (47 °C) independently by computing the dot product of the vectorized activation contrast image (βmap) with the NPS pattern of voxel weights (NPS-ωmap).

These individual values were then correlated with individual pain intensity ratings to evaluate the relationship between pattern of expression of the NPS and individual pain sensations. In addition, individual NPS pattern of expression were compared between stimulation conditions by performing a t-test. This analysis was performed in Matlab 2016a and Statistical Parametric Mapping Software (SPM) 12.

**Multivariate fMRI LASSO-PCR regression**. To further ensure that the lack of relationship between interindividual differences in pain sensitivity and its cerebral representations and given the fact that the NPS was developed to detect within-individual differences, we performed a whole-brain multivariate LASSO-PCR regression on our fMRI data using a similar approach as the one described in Wager et al.[21,36]. Compared to the NPS approach described above, this approach has two advantages: (1) it does not use a pre-defined map of activated areas and (2) the predictive algorithm is trained on our own dataset. For this purpose, we used the Canlab toolbox (https://canlab.github.io). We performed a five-fold nested cross-validation procedure, which was repeated five times. This means that at every repetition, one fifth of our dataset was used as test data and the rest was used as training data. The test data changed at each repetition, meaning that all data were included in the test set once. At each fold, lambda was estimated to minimize the mean squared error. In addition, we performed a bootstrap test with 5000 samples.

*Differences in brain activation between pain sensitivity classes in response to noxious heat.* To further investigate individual differences in brain activation associated with noxious heat stimulation, two group-level F-tests were performed to define any overall differences between the pain sensitivity classes. In addition, 6 group-level *t*-tests were performed to compare each class individually, using the following contrasts: High > Low Pain Sensitivity class, High < Low Pain Sensitivity class, Moderate > Low Pain Sensitivity class, Moderate < Low Pain Sensitivity class, High > Moderate Pain Sensitivity class, and High < Moderate Pain Sensitivity class.

*Effect of stimulus intensity.* To confirm that our paradigm was sensitive to changes in stimulus intensity, differences in brain activation between high intensity and low intensity stimuli were analyzed.

The univariate main analyses described above were replicated for each fMRI series using the low intensity stimuli as regressor of interest. Paired t-tests were then performed to investigate the effect of stimulus intensity on brain activation in the heat, cold, and auditory fMRI series. For each modality, this analysis included two steps: 1. paired t-tests of the high vs. low intensity stimuli at the single subject level; and 2. a group-level GLM analysis of the individual copes resulting from the previous step.

For all fMRI analyses, a clustering z threshold of 3.1 and p threshold of 0.05 were used.

*Analyses of potential confounds.* Kruskal–Wallis tests were completed to exclude further potential explanations of our main results, i.e., the lack of relationship between reported pain intensity and brain activation associated with high intensity heat stimulation.

*Effect of head motion.* The effect of pain sensitivity on head motion in the scanner during the heat paradigm was analyzed. Two motion parameters were used as dependent variables in separate Kruskal–Wallis analyses: Root Mean Squared (RMS), vector including estimated rotation and translation parameters, and frame-wise displacement (FWD), while pain sensitivity classes were used as factor in these analyses.

*Effect of repeated stimulation over time.* Given that each heat series included 7 stimuli and each series was distributed across the entire MRI session, we sought to determine if pain intensity rating (dependent variable) differed across repeated presentations and/or between classes of pain sensitivity (factors) by performing an additional Kruskal–Wallis test.

**Reporting summary**. Further information on research design is available in the Nature Research Reporting Summary linked to this article.

## Data availability

Data is available upon request. Source data are provided with this paper and activation maps have been uploaded to Github (https://github.com/coghill-painlab/IDP_fMRI_activationMaps). Source data are provided with this paper.

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

## Acknowledgements

We would like to thank Martin A. Garenfeld, Christian K. Mortensen, Victor J. 2nd Schneider, Gregory R. Lee, Blaise V. Jones, David L. Moore, Benjamin Hunter, and Catherine Jackson for their contribution to this study. This work is supported by the National Institute of Neurological Disorders and Stroke (R01 NS085391), the Serra Hunter Programme (MLS), and the National Insitutes of Health (NIH) Clinical and Translational Science Award (CTSA) program (2UL1TR001425; REDCap).

## Author contributions

Conceptualization, R.C.C.; Methodology, R.C.C., C.D.K., and H.N.A.; Investigation, H.N.A. and E.L; Software: M.E.H., W.A.H., and R.C.C.; Formal Analysis: M.E.H., M.L.S., and J.P.; Writing – Original Draft, M.E.H. and R.C.C.; Writing – Review & Editing, M.E.H., R.C.C., C.D.K., M.L.S., K.R.G., and H.N.A.; Funding Acquisition, R.C.C.; Resources, R.C.C.; Supervision, K.R.G.

## Competing interests

The authors declare no competing interests.
