## [Peer Review File · Nature Communications]

Dissociation between individual differences in self-reported pain intensity and underlying brain activationREVIEWER COMMENTS

Reviewer #1 (Remarks to the Author):

This paper tackles the issue of understanding individual differences in pain reporting to psychophysically constrained noxious thermal stimulation. This work builds on the heavily cited paper of the senior author from 2003. The rationale for this study is considered and clearly presented, detailing the importance of biomarkers for the development of novel therapeutics and for patients unable to provide conventional self-report. It is an important issue for the field and for that reason I would like to see it published. That said, I think there are considerable changes and additions to the manuscript required to achieve that.

In the main the writing is clear, but as a general comment, there is no road map for the analyses that are presented. The journal style places the methods at the end of the manuscript and so some additional statements in the introduction that give the reader some indication of what is to come would be welcome. As an example, there was no clue provided in the introduction that the NPS or any multivariate approaches would be used here and came as a surprise. Similarly, introductory statements to each analysis would improve the flow of the text. The results bounce back and forth from psychophysics to fMRI and I would suggest dealing with the psychophysical and demographic profile first, providing an in-depth characterisation of the cohort. The reporting of the different modalities (heat, cold, auditory) is unbalanced and in its current form, the description of cold pain does not seem to add much to the story. The bulk of the report is on noxious heat pain and I recommend more focus here (for reasons to follow) at the expense of the cold pain reporting. The auditory control condition does add to the story but is insufficiently considered in the discussion at present.

The sample size is large (>100 participants) and encompasses a relatively broad range of individuals. Rightly, both male and female participants are studied. Of note, adolescent respondents make up a proportion of the cohort and psychological profiles often excluded in healthy volunteer studies are included here. This reviewer does find it galling that 'middle age' stops at 44!

The characterisation of individuals of different sensitivity profiles using psychophysics is thorough and well described, based upon examination of VAS scale responses to noxious heat stimulation at a range of temperatures. Psychophysics is not my area of expertise and accordingly I will not comment on this part of the manuscript, but to my relatively untrained eye the approaches are considered and the outcomes of these analyses reasonable, as a means by which to stratify individuals into three classes of sensitivity. Rather than the provided scatterplots, however, I would prefer to see a histogram that demonstrates not only the range of scores but also their frequency; such a figure would help understand the distribution of VAS scores.

This is a big paper, but in summary: During fMRI, three different modalities (noxious heat, noxious cold, and non-noxious auditory stimulation) are studied, although the coverage of each modality differs considerably within the manuscript; noxious heat stimulation being by far the deeply investigated. The chosen (block-design) paradigms are straightforward in design and have been well-studied previously, but some of the results are surprising. Main effects of stimulation are identified in all classes, but regression analyses of BOLD responses to both classes of noxious stimulation, using subjective report as an explanatory variable, did not provide significant associations. The Neurological Pain Signature (NPS) of Wager et al. is also applied to noxious heat data as a form of multivariate regression, also unsuccessfully. By contrast, relationships were identified between subjectively reported indices of auditory stimulation in plausible regions of the brain. fMRI analyses comparing the three sensitivity classes did not show differences in the main effects of stimulation. Within-subject analyses examining differences between high and low heat stimulation were identified for thermal pain (although the fMRI results are not detailed other than in Figures 5B and C, which seems to be an oversight); subjective reports of cold pain did not differ between high and low levels of stimulation, but

differences were discernible using fMRI. Similar to cold pain, subjectively reported responses to auditory stimulation did not differ, but differences between high and low auditory stimulation were identified in the brain. The within-subject analyses provide a useful sanity check, but particularly in the case of thermal pain, it has been demonstrated countless times that fMRI is sensitive to detecting differences in temperature within an individual. As each person acts as their own control in these analyses, detecting effects is considerably more straightforward.

Importantly, these findings are contrary to the original report of Coghill et al., 2003, who in a much smaller sample identified a relationship between individual differences in pain reporting to a noxious thermal stimulation in S1, ACC and PFC, but not thalamus. These contemporary null findings are considered in terms of the utility of biomarker development for pain, although the discussion is limited in its scope much above a summary of the results. A wider and deeper consideration of why these findings differ from those published previously is encouraged.

In essence these null findings are what is important here, but I think the authors must attempt to provide more understanding about the different elements of inter-individual variability, in addition to differences in pain reporting, namely, age, gender, psychometric profile and potentially also their sensitivity classes. I am loathed to submerge the authors in endless new analyses, but some further experiments are likely to be insightful in understanding the variability in these individuals. There is sufficient data available to perform these investigations in various sub-strata of the larger cohort. Importantly, these additional analyses will enable the authors to state that null findings are not due to other factors additional to the reporting of their individualised pain experiences.

The assumptions of a linear VAS/BOLD relationship are important and should be further investigated. The psychophysical findings entertain the possibility that the relationship between VAS and BOLD response may alter between sensitivity classes. If we look within each sensitivity class separately, can VAS/BOLD relationships be identified then? A further question: is the broadened demographic and psychometric profile providing additional 'noise'? In particular, are further multiple linear regression analyses with additional explanatory psychometric variables, or stratification of the cohort according to these variables, insightful here? As the profiles do not differ across sensitivity classes, ANCOVA analyses would be permissible. How are the between-class ANOVA analyses altered with the addition of psychometric nuisance variables? Similarly, what is the effect of removing the adolescent participants? I wonder if, in a subset of the participants, with a profile as similar as possible to the 2003 report, a relationship may be found? This could be as simple as some a priori ROI analysis on regions previously shown to demonstrate associations. Should any of these analyses prove to identify a significant relationship, it would help to parameterise limits for putative biomarkers. I am choosing to ignore what the implications of this might be for cohorts of chronic pain patients, which are likely to be even more heterogeneous than the current cohort: and, of course, where such biomarkers are really needed. A comment on the translation of predictors of pain response to patients with persistent pain in the discussion is noted, but I would encourage further commentary, however speculative, on how we can further investigate why resting-state fMRI in highly heterogeneous chronic pain populations but not predict acute pain in otherwise pain-free volunteers. It should be noted that the recent Tonic Pain signature referenced offers at best a mixed performance when applied to back pain participants.

I am very interested in the white matter activations and also in the findings of deactivation in the main effects. I could not remember having seen reports of significant deactivations in the historical papers detailing responses to thermal stimulation (e.g. Derbyshire, 1997; Coghill, 1999 [both PET]; Brooks, 2002; Marquand, 2009, [both in Neuroimage]etc), nor in the historical reviews (e.g. Peyron, 2000, Clin Neurophysiol) and accordingly these strike me as an unusual finding. I think it unlikely that all of these early reports chose to only report positive brain activations in these papers, nor are they purely a consequence of the ability to detect smaller effects due to a larger sample size. White matter activations are also not present in the original reports and assuming they are not artifactual I think some further consideration of these findings is necessary; the recent paper of Gore et al (2019) in Mag Res Imaging may be a good starting point here.

I am also interested in the application of FSL-FIX. Although more commonly applied to resting-state

fMRI data, I do not have a theoretical issue with its use on task data, and the training of FIX on 'held out' data is entirely appropriate. FIX identifies and removes variance components identified in the training set as noise. Was there a particular reason why this approach was implemented here? I also note that intensity normalisation was also performed following application of FIX. This is a slightly non-standard step in first-level analysis and I would like to know the rationale for its inclusion. I would also like to know whether the FIX output for the adolescent participants differed from the other participants- were the same components being removed? Were the high-pain sensitive participants different in the components that were removed? Again, understanding any variability within the cohort is likely to be insightful.

I doubt that it is of relevance to the results of the univariate analyses, but I do wonder whether the removal of variance components affects the application of the NPS. Of course, pattern recognition techniques recognise features across the brain which may or not be related to brain activation. It was not clear to me whether or not the NPS was retrained on these data first or whether the pre-existing weights from the original report were used. Could these differing pre-processing strategies be important? NPS is a sparse representation, rather than utilising the whole brain dataset, but due to registration errors and partial voluming effects, the white matter activation may potentially be confounding here also.

More detail on the application of the NPS is necessary. It is not exactly clear here to me what the NPS response coefficients mean here, or whether we expect them to increase linearly across subjects; the original paper of 2013 if I recall correctly does not suggest this. While we can see that NPS coefficients differ between 'painfully hot' and 'even more painfully hot', as a sanity check one would expect decent classifier performance between these stimulus classes, as per the original paper. Demonstrating classification of pain vs rest and 'hot vs hotter' pain would demonstrate that the NPS is working as intended and be a useful inclusion to the paper.

The NPS was originally trained on individuals receiving noxious stimulation at multiple intensities. While the authors correctly state that multivariate approaches offer the opportunity of improved sensitivity to detect spatially distributed effects, I suspect that there are alternative multivariate regression pattern recognition approaches that might offer better performance (the early paper of Marquand et al in Neuroimage is one example of multivariate regression; I think the recent review of van der Meisen and Wager also discuss MVR also). With a dataset this size, it would be possible to hold out a proportion of the data as a test set and attempt to train a PR regression algorithm to detect individual differences rather than attempting to force the NPS into performing this role. This approach would also circumvent any potential issues associated with having applied FSL-FIX to the data. I encourage the authors to consider these approaches, or as a minimum discuss the utility of their application.

Minor points:

- The numbers of participants recruited and excluded do not seem to match throughout. Can some arithmetic checking be performed please?
- How can 17 high heat stimuli and 4 low stimuli be distributed across three runs?
- Was there any consideration of the phase of the menstrual cycle in female participants?
- Was time of day considered across participants?
- A ten second block is short. Why was this chosen?
- Please add detail on the second and third level analyses- fixed effects, flame etc. Knowing about the effects of within-run and across-run variance is an important detail.
- The supplementary analyses of time and motion lack sufficient detail know exactly what was done.
- Does the effect of time analysis provide the same result for the cold pain data? While the range of cold pain scores is reasonable, the mean response is very low, suggesting a skewed distribution. A histogram would be very useful here.
- The term 'massive univariate analysis' is unusual- typo?
- My interest is piqued by the reference to ASL and rsfMRI data. Is there anything in these datasets that predicts pain responses?

- Was the order of acquisition of each series (3x heat, 1 x cold, ASL, rsfMRI, structurals) the same in all participants or counter-balanced? Cold pain in particular can result in carry over effects.
- In light of the worthy efforts of Karen Davis et al and their reporting of the perils of 'forensic fMRI' for pain, is there still a market for these fMRI-based services?
- Surely what is important is the added value provided by fMRI rather than uniquely its ability to reproduce subjective reporting?

Reviewer #2 (Remarks to the Author):

This manuscript examined neural mechanisms underlying between-individual differences in subjective pain ratings, which is a revisit to the previous fMRI study led by Coghill and his colleagues (2003, PNAS), but with a larger sample size (N = 101). Although there were significant changes of fMRI activation in multiple classical pain-processing areas in response to high- and low-intensity noxious stimulation (and also the contrast between the two), the authors could not find any significant relationship between fMRI activation and subjective report of pain intensity, which is somewhat surprising and opposite the conclusion of previous studies of the authors' own research group. Additional analyses on confounding factors such as head motion further supported the robustness of their conclusion. Overall, the work is very clear and interesting, unique in its large sample size, and its conclusion seems to have a substantial implication in the field of pain research. Also, the authors' effort to be critical of their own previous works is admirable. I have a few comments about this manuscript, on which I elaborate below.

1. Univariate vs. Multivariate: The authors used either univariate analysis or a pre-defined multivariate signature (i.e., NPS) to examine the effect of a pain intensity regressor. However, there is a possibility that this "between-individual difference of pain sensitivity" is a globally distributed pattern in the brain, which may not be captured by examining a single brain region or voxel. Also, this pattern predictive of the between-individual difference of pain ratings can be different from NPS, which was mainly designed to predict the within-individual variation of pain ratings. For this reason, I'm wondering whether the authors can identify a multivariate model predictive of subjective pain intensity based on the current data.

2. Activation vs. Connectivity: Some previous studies tried to identify functional brain network patterns related to between-individual differences of pain sensitivity. For example, Spisak et al., (2020, Nat Comms) identified resting-state connectivity signature predictive of pain threshold, and Lee et al., (2021, Nat Med) identified task-related connectivity signature predictive of tonic pain intensity evoked by the same amount of hot sauce, across different individuals. All these two studies used the multivariate, whole-brain functional connectivity features for modeling, which makes me think that the inter-individual differences in subjective pain intensity might be represented as a globally distributed neural process (as the comment above). In this regard, I'm wondering whether the authors can elaborate more on the explanation of different results from fMRI activation and connectivity studies.

3. Within-individual variation: It seems that there is no information about the within-individual variation of pain ratings for the same stimulus intensity (49°C). Can the authors provide this information (e.g., adding error bar in Figure 1A)? Also, this study demonstrated the significant brain activation associated with within-individual differences in stimulus intensity, but did not examine whether there is any significant fMRI activation associated with within-individual differences in pain ratings for the same stimulus intensity. A previous study led by Woo and colleagues (2017, Nat Comms) identified an fMRI activation-based multivariate signature predictive of within-individual variation of pain ratings after controlling for stimulus intensity, but they did not develop or test signature onto a single stimulus intensity. Although I don't want to complicate the current paper by adding new analyses, I'm wondering if it is possible to examine the relationship between within-individual differences in pain ratings for the same stimulus intensity and brain activation.

4. Rating bias: In this study, the authors used tests of rating-based discrimination thresholds in the

QST session to support the argument that between-individual variation of pain intensity is not attributed to rating bias. First, for clarification purposes, I'm curious about the similarity between pain ratings of the QST session and those of the actual fMRI scanning session. Second, the 49°C-descending paradigm failed to show the differences of discrimination thresholds between three classes of pain sensitivity, suggesting that it is hard to confirm the absence of rating bias during 49°C heat stimulation, and the temperatures used for fMRI scanning were 48°C and 47°C, which are quite close to 49°C. This seems to weaken the authors' argument against the rating bias issue.

5. Head motion: For the supplementary analyses of head motion effect, did the authors really examine the relationship between "pain intensity" and "head motion"? (p. 33: "The effect of individually reported pain intensity on motion in the scanner during the heat paradigm was analyzed.")? It seems that they examined the differences of head motion between the three classes of pain sensitivity, because they used Kruskal-Wallis test which is a non-parametric version of one-way ANOVA and they actually described this in the discussion (p. 14: "First, there was no relationship between head motion and pain sensitivity classes."). However, the pain intensity regressors from the main analyses were defined based on individual-level, not on class-level. Thus, the comparison of motion parameters between classes does not seem to be the most principled way. How about trying (1) including motion effect as a covariate of no interest in the group-level GLM analyses, or (2) calculating the correlation coefficient between head motion parameters and pain ratings?

6. Clarification:

1) It seems that 6 high-intensity stimuli (48°C) and 1 low-intensity stimulus (47°C) were delivered for each session (pp. 21-22), but the total number of stimuli was described as 17 for high and 4 for low (p. 26). Because there are three sessions, I think this should be 18 for high and 3 for low. Can the authors clarify this?

2) p. 27: I'm wondering about the definition of "spurious class".

3) p. 27: For determining the number of classes of pain sensitivity, did the authors conduct model selection separately for pain intensity and unpleasantness?

Reviewer #3 (Remarks to the Author):

The manuscript of Coghill's group describes a novel study concerning the dissociation between individual differences in self-reported pain intensity and brain responses. I agree with the authors on the main conclusion that ratings of pain intensity and pain unpleasantness are reliable measures of the subjective experience of pain, while brain responses are not at the between-subject level. It should be noted that almost identical results were obtained in our unpublished data, which demonstrated that the reported results are reliable and could be replicated in different research groups. Therefore, I fully agree with the final statement of the paper that "pain is what the patient says it is" remains, and the use of fMRI as an objective measure to infer reported pain intensity for medico-legal purposes needs to be considered with great caution.

The whole study is well-designed, and the manuscript is well-written. I have a number of comments and questions that I hope are helpful to the authors in revising their work.

Introduction (lines 76-78). The authors hypothesized that individual differences in reported painful and non-painful sensations are associated with individual differences in brain activation. However, it's not clear how this research hypothesis was derived, especially for the non-painful sensations. It should be noted that the authors did not mention any research findings of the non-painful sensations in the Introduction.

Results (lines 94-102). The authors are suggested to clarify whether the reported correlation results were obtained at the within-subject level or the between-subject level, since comparing brain activations associated with intra- and inter-individual pain perception is the main topic of this study. For example, the relevant information about correlations should be made clear in the following sentences: "results of the covariance analysis detected no relationship between reported pain

intensity and brain activation associated with high intensity heat stimuli” and “Results of the correlation analysis between the NPS expression and reported pain intensity failed to show a significant association”.

Results (lines 98-102). Effect of reported pain intensity using the multivariate NPS. This analysis aimed to address the issue of low sensitivity inherent to the massive univariate analysis. However, the adopted analysis, i.e., to define the overlap of brain activation between the NPS and their data and to evaluate the correlation between the NPS expression and individual ratings of pain intensity, is not a multivariate pattern analysis, by definition. To increase the sensitivity of detection brain activation associated with inter-individual pain perception, a real multivariate pattern analysis (e.g., training an MVPA model based on a subset of subjects and testing the performance of the model on the remaining subjects) should be performed.

Results (lines 104-114). Pain sensitivity classes. The authors divided all subjects into three classes of pain sensitivity (Low, Moderate, and High Pain Sensitivity) and performed some statistical comparisons between the three classes. This procedure, grouping continuous pain rating data into a small number of classes, has been criticized because it does not reflect the natural distribution of the data and is often associated with “P-hacking”. For this reason, the authors are suggested to perform between-subject correlation analyses between ratings of pain perception and other variables, e.g., brain responses, demographic and psychological variables.

Results (lines 115-127). Pain sensitivity classes. Each class of pain sensitivity had a very distinctive stimulus-response curve, which is quite nonlinear according to the fitting curve in Supplementary figure 2. However, two linear fit parameters, slope and intercept, were used for the comparisons between the three classes. Obviously, the choice of the fit parameters is not optimal. Moreover, the description of the results is not very accurate, e.g., “The individuals from the Low Pain Sensitivity class had the lowest intercept (intensity: $\beta_0 = -9.79$)” and “the participants from the Moderate Pain Sensitivity class had low intercept (intensity: $\beta_0 = -10.02$)”. The lowest intercept is larger than in the low intercept.

Results (lines 135-151). Rating-based discrimination thresholds. It would also be more reasonable to perform a between-subject correlation analysis between ratings of pain perception and discrimination thresholds of intensity ratings.

Results (lines 192-200). Effect of stimulus intensity. The comparisons of pain perception ratings and brain responses between high intensity stimuli and low intensity stimuli are not entirely reasonable, as the number of trials with high intensity and low intensity stimuli was quite different (e.g., heat pain: 17 vs. 4).

Discussion (lines 256-257). Interindividual differences in subjective reports of pain. It is well known that pain perception could be greatly influenced by psychological factors, such as depression, anxiety, and catastrophizing. However, these influences could not be considered as rating bias.

Discussion (lines 279-284). Brain activation associated with noxious stimulation, but not individual differences in reported pain intensity. The authors claimed that the results of the massive univariate analysis were further confirmed by multivariate assessment of activation. This statement is not correct, as the authors did not perform the real multivariate pattern analysis.

Discussion (line 299). “the mean and range of pain intensity ratings between the two studies were similar ...”. It’s not clear which two studies the authors refer to.

Discussion. The dissociation between individual differences in pain perception and brain responses was also observed in EEG studies. For example, In Hu and Iannetti PNAS 2019, the majority part of laser-evoked brain responses was significantly correlated with pain perception at the within-subject level, but not at the between-subject level (Figure 2B in this paper). This observation was more clearly presented in the supplementary Figure 1 of the paper

(<https://www.pnas.org/content/pnas/suppl/2019/01/08/1812499116.DCSupplemental/pnas.1812499116.sapp.pdf>). The discussion of EEG findings would be important to demonstrate the reliability of the obtained results.

Discussion (lines 341-342). The statement that subjective reports of pain and objective measurements of nociception might provide complementary information on different aspects of pain may not be accurate and should be discussed with caution.

Discussion (lines 349-354). The findings in the present study are quite different from chronic pain, and the relevant discussion could be removed.

Methods (line 369). Participants. Out of the remaining 105 participants should be wrong. It should be 109 participants!

Methods (lines 440-456). QST session. Why were different stimulus durations used in familiarization (plateaued for 5 seconds) and fMRI training (10-second long stimuli) sessions? To demonstrate the stability across sessions, the authors are suggested to assess the test-retest reliability of pain perception using, for example, the correlation analysis.

Methods (lines 458-464). fMRI training stimuli. Why was the number of stimuli so different in different conditions, e.g., six high intensity and one low intensity for heat stimuli, four high intensity and one low intensity for cold stimuli, and five high intensity and two low intensity for auditory stimuli?

Methods (lines 476-535). Behavioral/psychological measures. Different questionnaires were used to collect data from adult and adolescent participants. However, it's not clear how this difference was considered in the data analysis? Did the authors remove the adolescent participants when assessing the relationship between pain perception and psychological factors?

Methods (lines 565-567). Block-design fMRI series. The number of trials for cold and auditory stimuli was not specified.

Methods (lines 577-581). Statistical analyses. Classes of pain sensitivity were obtained based on ratings of the 5-second stimulus acquired during the familiarization part of the QST session. Could the same or similar results be obtained when the analysis was performed on ratings of the 10-second stimulus acquired during the fMRI training session?

Methods (lines 650 & 654). Preprocessing of the MRI data. Data were split into 25 components by performing PICA. Why were 25 components considered? FIX classifier trained on fMRI heat series of 34 independent participants. It's not clear which 34 independent participants were used?

Methods (line 693). It's not clear why two group-level F-tests were performed?

Methods (line 709). For all fMRI analyses, a clustering z threshold of 3.1 and p threshold of 0.05 were used. Did the authors consider the multiple comparison problem?

September 9, 2021

NCOMMS-21-02776

Dissociation between individual differences in self-reported pain intensity and underlying brain activation

Dear reviewers,

Thank you for taking the time to review our manuscript, NCOMMS-21-02776. We found your reviews very helpful in pointing out issues to correct and refine, and we have revised the manuscript extensively.

Below we list the changes we made to the revised manuscript and respond to your concerns. These changes are highlighted in yellow in the manuscript. Your comments are written in italics and our detailed answers in regular font.

In addressing your concerns, we conducted 82 additional analyses. We were very careful to keep the focus of the paper on the relationship between individually perceived pain and underlying brain mechanisms. The dissociation between the two highlighted in this paper challenges long-standing scientific ideas based on single unit electrophysiology and neuroimaging. Accordingly, we seek to keep the emphasis of this manuscript on this important and novel finding.

Sincerely,

Marie-Eve Hoeppli, Ph.D.
Post-Doctoral Fellow

Robert C. Coghill, Ph.D.
Professor of Pediatrics
Director, Pediatric Pain Research Center

Reviewer #1:

This paper tackles the issue of understanding individual differences in pain reporting to psychophysically constrained noxious thermal stimulation. This work builds on the heavily cited paper of the senior author from 2003. The rationale for this study is considered and clearly presented, detailing the importance of biomarkers for the development of novel therapeutics and for patients unable to provide conventional self-report. It is an important issue for the field and for that reason I would like to see it published. That said, I think there are considerable changes and additions to the manuscript required to achieve that.

Author response (AR): We are grateful and thank the reviewer for his/her high appreciation of our research.

In the main the writing is clear, but as a general comment, there is no road map for the analyses that are presented. The journal style places the methods at the end of the manuscript and so some additional statements in the introduction that give the reader some indication of what is to come would be welcome. As an example, there was no clue provided in the introduction that the NPS or any multivariate approaches would be used here and came as a surprise.

AR: A paragraph has been added at the end of the introduction, which summarizes the analyses reported in this manuscript.

Similarly, introductory statements to each analysis would improve the flow of the text.

AR: introductory statements have now been added to each analysis' results.

The results bounce back and forth from psychophysics to fMRI and I would suggest dealing with the psychophysical and demographic profile first, providing an in-depth characterization of the cohort.

AR: As suggested by the reviewer, we now moved the main psychophysical results before the presentation of the fMRI results.

The reporting of the different modalities (heat, cold, auditory) is unbalanced and in its current form, the description of cold pain does not seem to add much to the story. The bulk of the report is on noxious heat pain, and I recommend more focus here (for reasons to follow) at the expense of the cold pain reporting. The auditory control condition does add to the story but is insufficiently considered in the discussion at present.

AR: We thank the reviewer for their suggestion to reweight the results from the cold and auditory modalities. As we now describe in the introduction, the cold modality is a control condition for the heat paradigm, and the auditory paradigm is a control for investigating individual differences in non-painful conditions. We have now expanded our discussion of the auditory results.

The sample size is large (>100 participants) and encompasses a relatively broad range of individuals. Rightly, both male and female participants are studied. Of note, adolescent respondents make up a proportion of the cohort and psychological profiles often excluded in healthy volunteer studies are included here. This reviewer does find it galling that 'middle age' stops at 44!

AR: We thank the reviewer for their appreciation of the inclusion of adolescents (and very young middle-aged participants)!

The characterization of individuals of different sensitivity profiles using psychophysics is thorough and well described, based upon examination of VAS scale responses to noxious heat stimulation at a range of temperatures. Psychophysics is not my area of expertise and accordingly I will not comment on this part of the manuscript, but to my relatively untrained eye the approaches are considered and the outcomes of these analyses reasonable, as a means by which to stratify individuals into three classes of sensitivity. Rather than the provided scatterplots, however, I would prefer to see a histogram that demonstrates not only the range of scores but also their frequency; such a figure would help understand the distribution of VAS scores.

AR: We thank the reviewer for his/her suggestion. We replaced the scatterplots by violin plots with a barplot in figures 2 and 3 and in supplementary figure 2. Violin plots provide the information on the distribution and density of the ratings of intensity. The addition of the barplot displays the mean and standard error of the ratings.

This is a big paper, but in summary: During fMRI, three different modalities (noxious heat, noxious cold, and non-noxious auditory stimulation) are studied, although the coverage of each modality differs considerably within the manuscript; noxious heat stimulation being by far the deeply investigated. The chosen (block-design) paradigms are straightforward in design and have been well-studied previously, but some of the results are surprising. Main effects of stimulation are identified in all classes, but regression analyses of BOLD responses to both classes of noxious stimulation, using subjective report as an explanatory variable, did not provide significant associations. The Neurological Pain Signature (NPS) of Wager et al. is also applied to noxious heat data as a form of multivariate regression, also unsuccessfully. By contrast, relationships were identified between subjectively reported indices of auditory stimulation in plausible regions of the brain. fMRI analyses comparing the three sensitivity classes did not show differences in the main effects of stimulation. Within-subject analyses examining differences between high and low heat stimulation were identified for thermal pain (although the fMRI results are not detailed other than in Figures 5B and C, which seems to be an oversight); subjective reports of cold pain did not differ between high and low levels of stimulation, but differences were discernible using fMRI. Similar to cold pain, subjectively reported responses to auditory stimulation did not differ, but differences between high and low auditory stimulation were identified in the brain. The within-subject analyses provide a useful sanity check, but particularly in the case of thermal pain, it has been demonstrated countless times that fMRI is sensitive to detecting differences in temperature within an individual. As each person acts as their own control in these analyses, detecting effects is considerably more straightforward.

Importantly, these findings are contrary to the original report of Coghill et al., 2003, who in a much smaller sample identified a relationship between individual differences in pain reporting to a noxious thermal stimulation in S1, ACC and PFC, but not thalamus. These contemporary null findings are considered in terms of the utility of biomarker development for pain, although the discussion is limited in its scope much above a summary of the results. A wider and deeper consideration of why these findings differ from those published previously is encouraged.

AR: As suggested by the reviewer, we added more details on the differences between the two studies and about why these differences are unlikely to be responsible for the lack of findings.

In essence these null findings are what is important here, but I think the authors must attempt to provide more understanding about the different elements of inter-individual variability, in addition to differences in pain reporting, namely, age, gender, psychometric profile and potentially also their sensitivity classes. I am loathed to submerge the authors in endless new analyses, but some further experiments are likely to be insightful in understanding the variability in these individuals. There is sufficient data available to perform these investigations in various sub-strata of the larger cohort. Importantly, these additional analyses will enable the authors to state that null findings are not due to other factors additional to the reporting of their individualised pain experiences.

The assumptions of a linear VAS/BOLD relationship are important and should be further investigated. The psychophysical findings entertain the possibility that the relationship between VAS and BOLD response may alter between sensitivity classes. If we look within each sensitivity class separately, can VAS/BOLD relationships be identified then?

AR: We thank the reviewer for the suggestion. We ran additional GLM analyses with pain intensity rating as covariate of interest within classes of pain sensitivity. These analyses showed no relationship between pain intensity ratings and brain activation associated with high intensity heat stimuli. These analyses, including methods and results, are now reported in the supplementary analyses as Supplementary Analysis 1.

A further question: is the broadened demographic and psychometric profile providing additional 'noise'? In particular, are further multiple linear regression analyses with additional explanatory psychometric variables, or stratification of the cohort according to these variables, insightful here? As the profiles do not differ across sensitivity classes, ANCOVA analyses would be permissible. How are the between-class ANOVA analyses altered with the addition of psychometric nuisance variables?

AR: A GLM analysis including all psychological and demographic data as well as motion parameters as covariates of no interest did not reveal any relationship between brain activation associated with high intensity heat stimuli and rating of pain intensity. This new result suggests that these variables did not impact on the relationship between brain activation associated with high intensity heat stimuli and individual pain intensity ratings and further support our main finding. These results are now reported in the supplementary analyses section under supplementary analysis 2.

The between classes ANOVA was performed as a confirmatory analysis for the main regression analysis. As the addition of covariates to the main regression did not reveal any further relationship between perceived pain intensity and brain activation, we chose not to further explore potential effects of these covariates in the between classes ANOVA.

Similarly, what is the effect of removing the adolescent participants?

AR: A GLM analysis without the adolescent participants yielded the same result as the main GLM analysis, i.e. no relationship between brain activation and pain intensity rating. Results of this analysis are now detailed in the supplementary analyses section, under supplementary analysis 3.

I wonder if, in a subset of the participants, with a profile as similar as possible to the 2003 report, a relationship may be found? This could be as simple as some a priori ROI analysis on regions previously shown to demonstrate associations.

AR: We thank the reviewer for his/her suggestion. We defined a subsample of our participants that were matched by sex, race, pain sensitivity, and averaged age to the participants of the 2003 study and ran a new GLM analysis on that sample. Results of this analysis further supported our main findings by showing no relevant relationship between inter-individual pain sensation and brain activation associated with high intensity heat stimuli. These results are now reported in the supplementary analyses section, under supplementary analysis 4.

Should any of these analyses prove to identify a significant relationship, it would help to parameterise limits for putative biomarkers. I am choosing to ignore what the implications of this might be for cohorts of chronic pain patients, which are likely to be even more heterogeneous than the current cohort: and, of course, where such biomarkers are really needed. A comment on the translation of predictors of pain response to patients with persistent pain in the discussion is noted, but I would encourage further commentary, however speculative, on how we can further investigate why resting-state fMRI in highly heterogeneous chronic pain populations but not predict acute pain in otherwise pain-free volunteers. It should be noted that the recent Tonic Pain signature referenced offers at best a mixed performance

AR: The additional analyses performed in accordance with these suggestions further support our main finding, i.e. they did not reveal any relationship between brain activation associated with high intensity heat stimuli and individually perceived pain intensity in any unique subgroup. Thus, heterogeneous responses across subgroups are unlikely to contribute to the absence of a relationship with individual differences.

The distinction between findings from resting-state fMRI connectivity frequently investigated in chronic pain populations versus BOLD task activation in pain-free individuals is perplexing. Some researchers (Barrosa, Branco, Apkarian Translational Research 2021) suggest that the mechanisms involved during chronic pain are remarkably distinct from those that process acute pain. This is an emerging and potentially controversial view. In the present investigation we are focusing on the surprising dissociation between acute pain ratings and brain activation, and we are hesitant with the limited space available to further comment about a distinction (acute vs. chronic pain) that is not the focus of our investigation.

I am very interested in the white matter activations and also in the findings of deactivation in the main effects. I could not remember having seen reports of significant deactivations in the historical papers detailing responses to thermal stimulation (e.g. Derbyshire, 1997; Coghill, 1999 [both PET]; Brooks, 2002; Marquand, 2009, [both in Neuroimage]etc), nor in the historical reviews (e.g Peyron, 2000, Clin Neurophysiol) and accordingly these strike me as an unusual finding. I think it unlikely that all of these

early reports chose to only report positive brain activations in these papers, nor are they purely a consequence of the ability to detect smaller effects due to a larger sample size. White matter activations are also not present in the original reports and assuming they are not artifactual I think some further consideration of these findings is necessary; the recent paper of Gore et al (2019) in Mag Res Imaging may be a good starting point here.

AR: We thank the reviewer for his/her interest in this finding. We have frequently reported regional grey matter deactivations with PET (Coghill et al. 1994, Coghill et al. 1999) as well as with fMRI. These consistently involve the posterior cingulate and precuneus and likely reflect deactivation of the default mode network. We would note that these deactivations also did not exhibit a relationship with perceived stimulus intensity and have added this to the results and discussion.

We have now added more discussion on the white matter deactivation in the supplementary discussion section: Task-related white matter deactivations are not very well known, nor studied, although we have frequently encountered them. It has been previously shown that changes in white matter activation can be associated with tasks, such as visual tasks (Gore & al., 2019; Gawryluk & al., 2014) or mindfulness (Zeidan & al., 2015; Zeidan & al., 2019). Although this is the first report of decreased white matter activation associated with pain, we have previously reported global cerebral blood flow changes associated with painful stimulation (Coghill & al., 1998). We speculate that the white matter deactivations may represent a consequence of a global change in signal intensity in combination with global signal normalization (done in all of our previous studies). Further studies are needed to confirm and deepen our understanding of this relationship.

I am also interested in the application of FSL-FIX. Although more commonly applied to resting-state fMRI data, I do not have a theoretical issue with its use on task data, and the training of FIX on 'held out' data is entirely appropriate. FIX identifies and removes variance components identified in the training set as noise. Was there a particular reason why this approach was implemented here?

AR: We thank the reviewer for this very relevant question. FIX was chosen to ensure an optimal signal retention in our data preprocessing. Following the lack of results in our main analysis, we investigated the effect of preprocessing on our data and noticed a better retention of signal using FIX compared to other denoising techniques, such as CompCor. We established the most appropriate denoising method for our data by running a comparison between FIX and CompCor and comparing those results to the results of analyses on a dataset preprocessed without any denoising algorithm. The details of this comparison will be reported in another article that is currently in preparation. For this reason, results of this comparison are not reported in the current manuscript.

I also note that intensity normalization was also performed following application of FIX. This is a slightly non-standard step in first-level analysis, and I would like to know the rationale for its inclusion.

AR: The intensity normalization is a step that we have consistently employed in all our PET and fMRI analyses. It is used to minimize the effect of global signal changes on our data.

I would also like to know whether the FIX output for the adolescent participants differed from the other participants- were the same components being removed? Were the high-pain sensitive participants

different in the components that were removed? Again, understanding any variability within the cohort is likely to be insightful.

AR: As mentioned in the article, our classifier was trained on an independent subsample of our participants and was consistently used on all the participants included in the analyses described in this manuscript. We note that data from 10 participants is considered appropriate to serve as training data for FIX. In our case, for the training data, we used 3 scans from each of the 34 participants. This exhaustive identification of potential noise features ensures that we had an appropriate representation of our test sample in our training cohort. Thus, our classifier was trained to remove components matching the same set of spatial and temporal features across our entire sample. We further note that one of the largest sources of noise would be head motion, and that head motion did not differ as a function of pain sensitivity. Finally, the utilization of a de-noising algorithm on has been shown to reduce the rate of false positive findings (Eklund et al., 2019).

I doubt that it is of relevance to the results of the univariate analyses, but I do wonder whether the removal of variance components affects the application of the NPS. Of course, pattern recognition techniques recognise features across the brain which may or not be related to brain activation. It was not clear to me whether or not the NPS was retrained on these data first or whether the pre-existing weights from the original report were used. Could these differing pre-processing strategies be important? NPS is a sparse representation, rather than utilising the whole brain dataset, but due to registration errors and partial voluming effects, the white matter activation may potentially be confounding here also.

AR: Although not reported in this manuscript, the NPS was also used on our data that were not preprocessed with FIX and no relationship between the NPS response and pain intensity rating was found. Therefore, we do not believe that the preprocessing with FIX impacted on these results.

The NPS was not retrained on our data, so the pre-existing weights were used.

The NPS showed that it was efficient at distinguishing brain activations in response to stimuli of different intensities. Therefore, we do not believe that the preprocessing, nor the fact that it was not retrained, nor the white matter deactivation affected the ability of the NPS to detect changes in brain activation associated with inter-individual pain intensity ratings.

More detail on the application of the NPS is necessary. It is not exactly clear here to me what the NPS response coefficients mean here, or whether we expect them to increase linearly across subjects; the original paper of 2013 if I recall correctly does not suggest this. While we can see that NPS coefficients differ between 'painfully hot' and 'even more painfully hot', as a sanity check one would expect decent classifier performance between these stimulus classes, as per the original paper. Demonstrating classification of pain vs rest and 'hot vs hotter' pain would demonstrate that the NPS is working as intended and be a useful inclusion to the paper. The NPS was originally trained on individuals receiving noxious stimulation at multiple intensities.

AR: We clarified the analyses including the NPS in the methods section of the manuscript by adding the following:

“For each participant, a scalar value representing their expression of the NPS in response to heat stimulation was computed. Only voxels included in the original NPS definition were included in this computation. This computation was performed for the high intensity heat stimuli (48°C) and for the low intensity heat stimuli (47°C) independently by computing the cross product of the vectorized activation contrast image (β map) with the NPS pattern of voxel weights (NPS- ω map).

These individual values were then correlated with individual pain intensity ratings to evaluate the relationship between pattern of expression of the NPS and individual pain sensations. In addition, individual NPS pattern of expression were compared between stimulation condition by performing a t-test.”

The results of the analysis comparing our two stimulation conditions using the NPS are reported in the results section and in figure 5. These results support the results obtained with the univariate analyses, showing a significant difference in the NPS pattern of expression between our two stimulation conditions. This supports that the NPS is working as intended.

While the authors correctly state that multivariate approaches offer the opportunity of improved sensitivity to detect spatially distributed effects, I suspect that there are alternative multivariate regression pattern recognition approaches that might offer better performance (the early paper of Marquand et al in Neuroimage is one example of multivariate regression; I think the recent review of van der Meisen and Wager also discuss MVR also). With a dataset this size, it would be possible to hold out a proportion of the data as a test set and attempt to train a PR regression algorithm to detect individual differences rather than attempting to force the NPS into performing this role. This approach would also circumvent any potential issues associated with having applied FSL-FIX to the data. I encourage the authors to consider these approaches, or as a minimum discuss the utility of their application.

AR: The dissociation of brain activation from perceived pain, but not underlying nociceptive activity strongly conflicts with decades of thinking based on single unit electrophysiology as well as neuroimaging. Accordingly, we seek to keep the emphasis of this manuscript on this important and novel finding. The main purpose of this paper was to test the hypothesis that brain activation was related to perceived pain intensity and thereby identify brain mechanisms that support individual differences in pain. The opposite - developing a brain signature to predict perceived pain intensity is an entirely different hypothesis that is beyond the scope of the present investigation. In addition, testing one multivariate model after another could be a never-ending process if it did not yield the expected results – and, if it did yield the desired result, one may question how biased the final product would be after a tremendous number of iterations.

Minor points:

- *The numbers of participants recruited and excluded do not seem to match throughout. Can some arithmetic checking be performed please?*

AR: this has been adjusted.

- *How can 17 high heat stimuli and 4 low stimuli be distributed across three runs?*

AR: The distribution of the stimuli throughout the runs has been detailed in the methods. Series 1 and 2 included 6 high and 1 low intensity stimuli each, while series 3 included 5 high and 2 low intensity stimuli, for a total of 17 high and 4 low intensity stimuli.

- *Was there any consideration of the phase of the menstrual cycle in female participants?*

AR: There was no consideration of the menstrual cycle phase in female participants. A calculation of the standard deviation of pain intensity ratings shows very similar standard deviation values between male (sd = 3.08) and female (sd = 2.93), suggesting that menstrual phase contributed minimally to variability in female participants.

- *Was time of day considered across participants?*

AR: The time of day was not considered across participants. Given the fact that participants were tested throughout the day, such effect should be counterbalanced between participants and therefore not bias the results.

- *A ten second block is short. Why was this chosen?*

AR: Although a ten second block is short, it was sufficient to elicit pain sensations similar to the ones obtained with longer duration and described in Coghill et al. (2003) and to elicit significant brain activation in response to these stimuli. Therefore, we believe this duration to be sufficient for our purpose. Furthermore, longer duration stimuli would have been extremely difficult to tolerate for high pain sensitivity individuals and complicate their inclusion in this study.

- *Please add detail on the second and third level analyses- fixed effects, flame etc. Knowing about the effects of within-run and across-run variance is an important detail.*

AR: It is now detailed in the methods section that fixed effect was used in the second level and mixed-effect FLAME 1 and 2 was used in the third/group level analysis.

- *The supplementary analyses of time and motion lack sufficient detail know exactly what was done.*

AR: It is now clarified in the methods that two Kruskal-Wallis tests were performed. In the first test, differences in motion parameters (dependent variables: RMS and FWD) were tested between classes of pain sensitivity (factor). In the second one, differences in pain intensity ratings (dependent variable) were tested across all heat stimuli and between pain sensitivity classes (factors).

- *Does the effect of time analysis provide the same result for the cold pain data? While the range of cold pain scores is reasonable, the mean response is very low, suggesting a skewed distribution. A histogram would be very useful here.*

AR: The repetition of the analysis of the effect of time on the cold data yielded no significant results. Given the low number of stimuli included in the cold paradigm (only 4 high intensity cold stimuli), we do not believe this analysis to be very reliable and did not include these results in the manuscript. Information on the distribution of the cold ratings can be found in the violin plot included in supplementary figure 2.

- *The term 'massive univariate analysis' is unusual- typo?*

AR: This is an emerging term that is used to describe traditional GLM based analyses that are conducted independently for each of the voxels in the brain and then corrected for multiple comparisons using Gaussian random field theory or other methods. It is frequently used in contrast with multivariate approaches.

- *My interest is piqued by the reference to ASL and rsfMRI data. Is there anything in these datasets that predicts pain responses?*

AR: The analyses of ASL and rs-fMRI data are ongoing. These results will be reported in future publications. Here we focus exclusively on the activation data to test the hypothesis that brain activity is related to perceived pain intensity.

- *Was the order of acquisition of each series (3x heat, 1 x cold, ASL, rsfMRI, structurals) the same in all participants or counter-balanced? Cold pain in particular can result in carry over effects.*

AR: The order of the sequences is now detailed in the methods section of the manuscript: the structural volume was always acquired first. The BOLD and ASL sequences were then counter-balanced between participants. Cold stimuli were delivered in separate series and required swapping of the probe, thus ample time between series minimized concerns about carryover effects.

- *In light of the worthy efforts of Karen Davis et al and their reporting of the perils of 'forensic fMRI' for pain, is there still a market for these fMRI-based services?*

AR: There is increasing awareness among the scientific community about the risks of 'forensic fMRI' for pain. However, our work with patients has highlighted how complicated it is for them to get insurance coverage for their treatments in big part because of the lack of objective markers. We feel that markers for a specific pain syndrome (e.g. CRPS, fibromyalgia etc.) could be of potential utility for providing a mechanism-based diagnosis without running the risk of denying a patient's pain.

- *Surely what is important is the added value provided by fMRI rather than uniquely its ability to reproduce subjective reporting?*

AR: We thank the reviewer for this thought-provoking comment. We have directly witnessed in our work with patients the added clinical value of quantitative sensory testing and are convinced that fMRI could add to this. We think that more brain imaging-based studies are needed to clearly define the added value of fMRI to clinical diagnosis and treatment of chronic pain conditions.

Reviewer #2:

This manuscript examined neural mechanisms underlying between-individual differences in subjective pain ratings, which is a revisit to the previous fMRI study led by Coghill and his colleagues (2003, PNAS), but with a larger sample size (N = 101). Although there were significant changes of fMRI activation in multiple classical pain-processing areas in response to high- and low-intensity noxious stimulation (and

also the contrast between the two), the authors could not find any significant relationship between fMRI activation and subjective report of pain intensity, which is somewhat surprising and opposite the conclusion of previous studies of the authors' own research group. Additional analyses on confounding factors such as head motion further supported the robustness of their conclusion. Overall, the work is very clear and interesting, unique in its large sample size, and its conclusion seems to have a substantial implication in the field of pain research. Also, the authors' effort to be critical of their own previous works is admirable. I have a few comments about this manuscript, on which I elaborate below.

- 1. Univariate vs. Multivariate: The authors used either univariate analysis or a pre-defined multivariate signature (i.e., NPS) to examine the effect of a pain intensity regressor. However, there is a possibility that this "between-individual difference of pain sensitivity" is a globally distributed pattern in the brain, which may not be captured by examining a single brain region or voxel. Also, this pattern predictive of the between-individual difference of pain ratings can be different from NPS, which was mainly designed to predict the within-individual variation of pain ratings. For this reason, I'm wondering whether the authors can identify a multivariate model predictive of subjective pain intensity based on the current data.*

AR: We thank the reviewer for taking the time to review our paper and for this insightful comment. We agree that the limited brain areas included in the NPS can hinder findings throughout the brain. However, our univariate analyses showed significant brain activation in these areas, which should allow a reliable detection of these.

As we noted in our response to reviewer 1, the dissociation of brain activation from perceived pain, but not underlying nociceptive activity strongly conflicts with decades of thinking based on single unit electrophysiology as well as neuroimaging. Accordingly, we seek to keep the emphasis of this manuscript on this important and novel finding. The main purpose of this paper was to test the hypothesis that brain activation was related to perceived pain intensity and thereby identify brain mechanisms that support individual differences in pain. The opposite - developing a brain signature to predict perceived pain intensity is an entirely different hypothesis that is beyond the scope of the present investigation. In addition, testing one multivariate model after another could be a never-ending process if it did not yield the expected results – and, if it did yield the desired result, one may question how biased the final product would be after a tremendous number of iterations.

- 2. Activation vs. Connectivity: Some previous studies tried to identify functional brain network patterns related to between-individual differences of pain sensitivity. For example, Spisak et al., (2020, Nat Comms) identified resting-state connectivity signature predictive of pain threshold, and Lee et al., (2021, Nat Med) identified task-related connectivity signature predictive of tonic pain intensity evoked by the same amount of hot sauce, across different individuals. All these two studies used the multivariate, whole-brain functional connectivity features for modeling, which makes me think that the inter-individual differences in subjective pain intensity might be represented as a globally distributed neural process (as the comment above). In this regard, I'm wondering whether the authors can elaborate more on the explanation of different results from fMRI activation and connectivity studies.*

AR: We thank the reviewer for this interesting point. It likely deserves a full paper by itself as it really involves fundamental questions about the neural substrates of consciousness as well as the conscious experience of pain. In the discussion, we have now elaborated on the possibility that the instantiation of a conscious experience of pain may arise from a globally distributed process and indicate how this would represent a radical transformation of afferent input.

- 3. Within-individual variation: It seems that there is no information about the within-individual variation of pain ratings for the same stimulus intensity (49°C). Can the authors provide this information (e.g., adding error bar in Figure 1A)? Also, this study demonstrated the significant brain activation associated with within-individual differences in stimulus intensity but did not examine whether there is any significant fMRI activation associated with within-individual differences in pain ratings for the same stimulus intensity. A previous study led by Woo and colleagues (2017, Nat Comms) identified an fMRI activation-based multivariate signature predictive of within-individual variation of pain ratings after controlling for stimulus intensity, but they did not develop or test signature onto a single stimulus intensity. Although I don't want to complicate the current paper by adding new analyses, I'm wondering if it is possible to examine the relationship between within-individual differences in pain ratings for the same stimulus intensity and brain activation.*

AR: Within-individual variation in pain ratings is now displayed in supplementary figure 2. In addition, our results show that there is no effect of time on pain intensity ratings and that the overall ratings were very stable.

Although the question of the relationship of brain activation with within-individual differences in pain ratings at a fixed stimulus intensity is an interesting question, it is not the focus of this paper. The focus of this paper is on inter-individual differences in pain and their association with brain responses to highly noxious stimuli. The analyses on within-individual differences included in this manuscript served solely as analyses to confirm the power of our paradigms to detect changes in brain activation associated with small changes in perception. We do feel that the paper is highly complicated with its present focus and adding an additional focus would distract the reader from our main and crucially important finding.

- 4. Rating bias: In this study, the authors used tests of rating-based discrimination thresholds in the QST session to support the argument that between-individual variation of pain intensity is not attributed to rating bias. First, for clarification purposes, I'm curious about the similarity between pain ratings of the QST session and those of the actual fMRI scanning session.*

AR: We performed a Bland-Altman analysis to assess the test-retest reliability of pain perception in our study. The results of this analysis show an averaged difference of 1.5 VAS units between the two sessions, with ratings in the first session being typically greater than ratings in the second session. This is well illustrated by the plot below. This systematic bias might be related to exposure effect or to being in the MRI during the second session. Given that our main fMRI results used ratings acquired during the MRI session and our main psychophysical results used ratings acquired during the QST session, we do

not believe this change in pain perception affects our findings or our conclusions. These results are now reported in the supplementary analysis section under Supplementary analysis 5.

Second, the 49°C-descending paradigm failed to show the differences of discrimination thresholds between three classes of pain sensitivity, suggesting that it is hard to confirm the absence of rating bias during 49°C heat stimulation, and the temperatures used for fMRI scanning were 48°C and 47°C, which are quite close to 49°C. This seems to weaken the authors' argument against the rating bias issue.

AR: The discrimination analysis was performed on the 5s duration stimuli during the QST session. The ratings of these briefer, yet more intense stimuli than used in the scanner fell into approximately the same range (mean of 49°C stimuli: 4.88 vs. mean of 48°C: 3.92).

It is important to note that we did detect significant group differences in unpleasantness ratings during the 49°C descending paradigm. Although the results for the intensity ratings failed to achieve statistical significance, qualitative observation of the results still showed a trend towards difference in the 49°C discrimination between the low/moderate groups and the high sensitivity group: the high sensitivity group requires a greater decrease in temperature to report a change in pain intensity. Thus, these findings support our argument that rating biases were not wholly responsible for individual differences in pain ratings.

5. *Head motion: For the supplementary analyses of head motion effect, did the authors really examine the relationship between “pain intensity” and “head motion”? (p. 33: “The effect of individually reported pain intensity on motion in the scanner during the heat paradigm was analyzed.”)? It seems that they examined the differences of head motion between the three classes of pain sensitivity, because they used Kruskal-Wallis test which is a non-parametric version of one-way ANOVA and they actually described this in the discussion (p. 14: “First, there was no relationship between head motion and pain sensitivity classes.”). However, the pain intensity regressors from the main analyses were defined based on individual-level, not on class-level. Thus, the comparison of motion parameters between classes does not seem to be the most principled way. How about trying (1) including motion effect as a covariate of no interest in the group-level GLM analyses, or (2) calculating the correlation coefficient between head motion parameters and pain ratings?*

AR: We thank the reviewer for his/her comment and clarified throughout that the analysis on the head motion parameters was performed using the pain sensitivity classes. We also performed a replication of the main GLM analysis including the motion parameters as covariate of no interest. The results of this analysis further supported our findings, i.e. lack of relationship between pain intensity ratings and brain activation associated with highly noxious stimuli. Finally, we calculated the correlation between pain intensity ratings and head motion parameters. These results show no significant correlation between the two, further supporting the results presented in our main manuscript. Results of these additional analyses are reported in the supplementary analyses section of the manuscript under Supplementary analysis 9.

6. *Clarification:*

- 1) *It seems that 6 high-intensity stimuli (48°C) and 1 low-intensity stimulus (47°C) were delivered for each session (pp. 21-22), but the total number of stimuli was described as 17 for high and 4 for low (p. 26). Because there are three sessions, I think this should be 18 for high and 3 for low. Can the authors clarify this?*

AR: We thank the reviewer for his/her comment and clarified the repartition of our 17 high and 4 low stimuli across the three fMRI series (Series 1 and 2 included 6 high and 1 low intensity stimuli each, while series 3 included 5 high and 2 low intensity stimuli, for a total of 17 high and 4 low intensity stimuli).

- 2) *p. 27: I’m wondering about the definition of “spurious class”.*

By ‘spurious’, we mean that this group does not reflect the existence of a true population subgroup and more likely reflects an over-extraction of mixture classes that lacks parsimony .

- 3) *p. 27: For determining the number of classes of pain sensitivity, did the authors conduct model selection separately for pain intensity and unpleasantness?*

AR: Each model included both pain intensity and pain unpleasantness rating, hence the selection was based on both types of ratings. This has been clarified in the methods section of the paper.

Reviewer #3:

The manuscript of Coghill's group describes a novel study concerning the dissociation between individual differences in self-reported pain intensity and brain responses. I agree with the authors on the main conclusion that ratings of pain intensity and pain unpleasantness are reliable measures of the subjective experience of pain, while brain responses are not at the between-subject level. It should be noted that almost identical results were obtained in our unpublished data, which demonstrated that the reported results are reliable and could be replicated in different research groups. Therefore, I fully agree with the final statement of the paper that "pain is what the patient says it is" remains, and the use of fMRI as an objective measure to infer reported pain intensity for medico-legal purposes needs to be considered with great caution.

The whole study is well-designed, and the manuscript is well-written. I have a number of comments and questions that I hope are helpful to the authors in revising their work.

AR: We thank the reviewer for their support and enthusiasm for our paper. We hope to answer their comments to their satisfaction below.

Introduction (lines 76-78). The authors hypothesized that individual differences in reported painful and non-painful sensations are associated with individual differences in brain activation. However, it's not clear how this research hypothesis was derived, especially for the non-painful sensations. It should be noted that the authors did not mention any research findings of the non-painful sensations in the Introduction.

AR: We thank the reviewer for highlighting this imbalance in our introduction. We now clarified that the investigation of the cold pain paradigm was to determine if the heat findings would generalize to a different noxious stimulus, and that the auditory paradigm served as a innocuous control condition for the noxious stimuli.

Results (lines 94-102). The authors are suggested to clarify whether the reported correlation results were obtained at the within-subject level or the between-subject level, since comparing brain activations associated with intra- and inter-individual pain perception is the main topic of this study. For example, the relevant information about correlations should be made clear in the following sentences: "results of the covariance analysis detected no relationship between reported pain intensity and brain activation associated with high intensity heat stimuli" and "Results of the correlation analysis between the NPS expression and reported pain intensity failed to show a significant association".

AR: We thank the reviewer for this comment. We have clarified this throughout the manuscript.

Results (lines 98-102). Effect of reported pain intensity using the multivariate NPS. This analysis aimed to address the issue of low sensitivity inherent to the massive univariate analysis. However, the adopted analysis, i.e., to define the overlap of brain activation between the NPS and their data and to evaluate the correlation between the NPS expression and individual ratings of pain intensity, is not a multivariate pattern analysis, by definition. To increase the sensitivity of detection brain activation associated with inter-individual pain perception, a real multivariate pattern analysis (e.g., training an MVPA model based

on a subset of subjects and testing the performance of the model on the remaining subjects) should be performed.

AR: We thank the reviewer for this insightful comment. We believe that the NPS should have sufficient power of detection to reveal the investigated effect. As noted above, the NPS was highly sensitive to within-subjects differences, so it was performing as expected. We have now clarified that the NPS was derived from a multivariate analysis.

As we noted in our response to reviewer 1, the dissociation of brain activation from perceived pain, but not underlying nociceptive activity strongly conflicts with decades of thinking based on single unit electrophysiology as well as neuroimaging. Accordingly, we seek to keep the emphasis of this manuscript on this important and novel finding. The main purpose of this paper was to test the hypothesis that brain activation was related to perceived pain intensity and thereby identify brain mechanisms that support individual differences in pain. The opposite - developing a brain signature to predict perceived pain intensity is an entirely different hypothesis that is beyond the scope of the present investigation. In addition, testing one multivariate model after another could be a never-ending process if it did not yield the expected results – and, if it did yield the desired result, one may question how biased the final product would be after a tremendous number of iterations.

Results (lines 104-114). Pain sensitivity classes. The authors divided all subjects into three classes of pain sensitivity (Low, Moderate, and High Pain Sensitivity) and performed some statistical comparisons between the three classes. This procedure, grouping continuous pain rating data into a small number of classes, has been criticized because it does not reflect the natural distribution of the data and is often associated with “P-hacking”. For this reason, the authors are suggested to perform between-subject correlation analyses between ratings of pain perception and other variables, e.g., brain responses, demographic and psychological variables.

AR: Our primary analysis was indeed a regression analysis of inter-individual pain ratings with brain activation.

Instead of arbitrarily splitting participants into subgroups for further analyses, we chose an objective procedure – mixture modeling – to subgroup participants. This objective procedure does in fact reflect the natural distribution of the data.

Moreover, the subgroups analysis was confirmatory in nature – to confirm that highly sensitive individuals had no differences in brain activation from insensitive individuals and to confirm that individuals discriminated pain in a fashion reflecting their sensitivity. As the analysis of brain activation did not reveal differences between groups, there should be no concerns about p-hacking.

Finally, we performed inter-individual correlations between pain intensity and unpleasantness ratings and psychological variables. Three additional statistical tests were performed to investigate differences in pain ratings within demographic variables: 1) a t-test was performed to investigate differences in pain ratings between sex; 2) an ANOVA was performed to investigate differences in pain ratings between

aces; 3) a correlation was performed to investigate the relationship between pain ratings and age. Results did not show any correlations between pain ratings and psychological variables. Similarly, no differences were found in pain ratings between sex or between races and no relationship was found between pain ratings and age. These results further support our main findings. These results have been added to the supplementary analyses section of the manuscript, including tables summarizing them, under Supplementary analysis 6 and 7.

Results (lines 115-127). Pain sensitivity classes. Each class of pain sensitivity had a very distinctive stimulus-response curve, which is quite nonlinear according to the fitting curve in Supplementary figure 2. However, two linear fit parameters, slope and intercept, were used for the comparisons between the three classes. Obviously, the choice of the fit parameters is not optimal. Moreover, the description of the results is not very accurate, e.g., “The individuals from the Low Pain Sensitivity class had the lowest intercept (intensity: 60 = -9.79)” and “the participants from the Moderate Pain Sensitivity class had low intercept (intensity: 60 = -10.02)”. The lowest intercept is larger than in the low intercept.

AR: We now realize that we did not adequately describe the line fitting procedure. These curves were fit to a power function according to the classic Stevens Power Law. All curve fits were significant, and statistics are now included. To further clarify, we now refer to the proportionality constant and exponent according to Stevens Power Law.

Results (lines 135-151). Rating-based discrimination thresholds. It would also be more reasonable to perform a between-subject correlation analysis between ratings of pain perception and discrimination thresholds of intensity ratings.

AR: Although our approach using pain sensitivity classes defined by a mixture model analysis is an objective approach, we ran Pearson’s correlations to further investigate the relationship between pain ratings and discrimination thresholds and confirm that our approach using classes of pain sensitivity did not affect our results.

The results revealed significant correlations between pain intensity and 43-ascending discrimination thresholds for pain intensity ($r(99) = -0.26$, $p = 0.007$), which remained significant after Bonferroni correction. In addition, significant correlations were found between pain unpleasantness and 43-ascending discrimination thresholds for unpleasantness ($r(99) = -0.39$, $p = 0.00006$), which also remained significant after Bonferroni correction.

The relationship between pain unpleasantness and 49-descending discrimination thresholds was significant for unpleasantness ratings ($r(99) = -0.24$, $p = 0.015$), but this did not survive Bonferroni correction.

These correlation results support the findings of our approach and further suggest that individuals vary in their ability to discriminate noxious stimuli based on their sensitivity to pain. These results have been added to the supplementary analyses section of the manuscript under Supplementary analysis 8, including tables summarizing these results.

Results (lines 192-200). Effect of stimulus intensity. The comparisons of pain perception ratings and brain responses between high intensity stimuli and low intensity stimuli are not entirely reasonable, as the number of trials with high intensity and low intensity stimuli was quite different (e.g., heat pain: 17 vs. 4).

AR: Despite the difference in the number of stimuli, we believe that this finding is still valid for the following reasons: 1) despite a low number of low intensity heat stimuli, significant brain activation was revealed in response to these stimuli. 2) similar findings were established in our two other paradigms that have less differences in the number of stimuli. 3) The family of General Linear Model analyses is robust to unbalanced designs, as long as the same number of participants are tested in each condition, which is our case. Moreover, the within-subjects stimulus response data is included to primarily confirm that our imaging and stimulation paradigms are sufficiently sensitive to detect small differences in brain activation.

Discussion (lines 256-257). Interindividual differences in subjective reports of pain. It is well known that pain perception could be greatly influenced by psychological factors, such as depression, anxiety, and catastrophizing. However, these influences could not be considered as rating bias.

AR: We certainly agree that the above factors hold the potential to influence pain itself. However, we were referring to the rating process, such that the reporting and communication of the pain experience could be systematically altered by these factors. This has been clarified in the manuscript.

Discussion (lines 279-284). Brain activation associated with noxious stimulation, but not individual differences in reported pain intensity. The authors claimed that the results of the massive univariate analysis were further confirmed by multivariate assessment of activation. This statement is not correct, as the authors did not perform the real multivariate pattern analysis.

AR: We have now clarified that the NPS it is not a MVPA, but it was developed from one.

Discussion (line 299). "the mean and range of pain intensity ratings between the two studies were similar ... ". It's not clear which two studies the authors refer to.

AR: It is now clarified that these studies are the current one and the one from Coghill et al. (2003).

Discussion. The dissociation between individual differences in pain perception and brain responses was also observed in EEG studies. For example, In Hu and Iannetti PNAS 2019, the majority part of laser-evoked brain responses was significantly correlated with pain perception at the within-subject level, but not at the between-subject level (Figure 2B in this paper). This observation was more clearly presented in the supplementary Figure 1 of the paper (<https://www.pnas.org/content/pnas/suppl/2019/01/08/1812499116.DCSupplemental/pnas.1812499116.sapp.pdf>). The discussion of EEG findings would be important to demonstrate the reliability of the obtained results.

AR: We thank the reviewer on this very interesting and relevant article that is in line with our current findings. We added a paragraph discussing this article to the discussion section of the manuscript:

“Interestingly, this disconnect between the cerebral representations of intra- and inter-individual pain perception has been previously observed and reported using EEG (Hu & Iannetti, 2019, PNAS). In that EEG study, within-individual differences in nociception were associated with changes in EEG responses, while inter-individual differences in pain are not associated with changes in EEG responses. These previous findings further support our present findings. These results taken together suggest that these brain activations might be more associated with nociception than with individual differences in perceived pain.”

Discussion (lines 341-342). The statement that subjective reports of pain and objective measurements of nociception might provide complementary information on different aspects of pain may not be accurate and should be discussed with caution.

AR: This statement has now been replaced by the following statement: “These results taken together suggest that cerebral representations of pain might be more associated with nociception than with individual perception.”

Discussion (lines 349-354). The findings in the present study are quite different from chronic pain, and the relevant discussion could be removed.

AR: It is common to first define cerebral representation of pain and/or brain biomarkers for pain in acute pain before testing it in chronic pain conditions. This is a sensitive approach given that cerebral representations of acute and chronic pain are often reported in similar brain areas. Therefore, we do not believe that recommending cautiousness when using brain biomarkers in chronic pain conditions is out of the scope of this paper.

Methods (line 369). Participants. Out of the remaining 105 participants should be wrong. It should be 109 participants!

AR: we thank the reviewer for this observation and corrected our arithmetic.

Methods (lines 440-456). QST session. Why were different stimulus durations used in familiarization (plateaued for 5 seconds) and fMRI training (10-second long stimuli) sessions? To demonstrate the stability across sessions, the authors are suggested to assess the test-retest reliability of pain perception using, for example, the correlation analysis.

AR: We have used 5-second long stimuli for training since 1991, this standardized training procedure allows to generalize our findings across studies. Our lab’s recent studies used 10-second long stimuli to ensure adequate MRI signal.

As reported in reviewer #2’s comments, we performed a Bland-Altman analysis to assess the test-retest reliability of pain perception in our study. This was done on the fMRI training stimuli during the QST session and the stimuli from the fMRI session (both with equal temperatures and 10s plateau durations). The results of this analysis show an averaged difference of 1.5 VAS unit between the two sessions, with ratings in the first session being typically greater than ratings in the second session. This is well illustrated by the plot below. This systematic bias might be related to exposure effect or to being in the MRI during the second session. Given that our main results used ratings acquired during the MRI

session, we do not believe this change in pain perception affects our findings or our conclusions. These results are now reported in supplementary analyses section under Supplementary analysis 5.

Methods (lines 458-464). fMRI training stimuli. Why was the number of stimuli so different in different conditions, e.g., six high intensity and one low intensity for heat stimuli, four high intensity and one low intensity for cold stimuli, and five high intensity and two low intensity for auditory stimuli?

AR: As it is now clarified in the methods section of the manuscript, the first and second heat series included 6 high and 1 low intensity stimuli, while the last one included 5 high and 2 low intensity stimuli. Since there was only one auditory series, we decided to replicate the number of stimuli of the last heat run, to have a higher number of low intensity stimuli (2). This was decided to increase our statistical power when performing analyses on the low intensity conditions. The number of cold stimuli was reduced from 7 to 5 during the pilot phase of this study for technical reasons: the stimulator required significant time to cool down to reach our target temperatures, preventing us from having the same number of stimuli as in the other conditions.

Methods (lines 476-535). Behavioral/psychological measures. Different questionnaires were used to collect data from adult and adolescent participants. However, it's not clear how this difference was

considered in the data analysis? Did the authors remove the adolescent participants when assessing the relationship between pain perception and psychological factors?

AR: The adolescents were included in the analyses, unless specified otherwise. When psychological factors were analyzed, the appropriate questionnaires were used to assess adolescents.

Methods (lines 565-567). Block-design fMRI series. The number of trials for cold and auditory stimuli was not specified.

AR: This has now been added to the methods section of the manuscript.

Methods (lines 577-581). Statistical analyses. Classes of pain sensitivity were obtained based on ratings of the 5-second stimulus acquired during the familiarization part of the QST session. Could the same or similar results be obtained when the analysis was performed on ratings of the 10-second stimulus acquired during the fMRI training session?

AR: The definition of classes of pain sensitivity using mixture model analysis could not be performed using the 10-second stimuli, given that only two temperatures were used in this paradigm, i.e. 47 and 48. Such an analysis would not provide a complete picture of pain sensitivity as did the one on the 5-second stimuli, which included 8 temperatures ranging from 35 to 49°C.

Methods (lines 650 & 654). Preprocessing of the MRI data. Data were split into 25 components by performing PICA. Why were 25 components considered? FIX classifier trained on fMRI heat series of 34 independent participants. It's not clear which 34 independent participants were used?

AR: Typically, researchers define a number of ICA components to extract between 20 and 30. We ran some preliminary analyses using 20, 25, and 30 components to define the number of components to extract from our datasets to optimize the FIX classifier. 25 components allowed a good definition of these components. The 34 independent participants were acquired after the 101 participants used in these analyses and underwent identical recruitment, stimulation, and imaging procedures.

Methods (line 693). It's not clear why two group-level F-tests were performed?

AR: The F-tests allow more global investigations of differences between groups than the t-tests. This has now been clarified in the methods section of the manuscript.

Methods (line 709). For all fMRI analyses, a clustering z threshold of 3.1 and p threshold of 0.05 were used. Did the authors consider the multiple comparison problem?

AR: The post-stat steps in FSL include corrections for multiple comparisons. Hence, no additional correction was applied to the fMRI data post-hoc.

REVIEWER COMMENTS

Reviewer #1 (Remarks to the Author):

Response to resubmission- Hoeppli et al

I thank the authors for their detailed responses to my comments and those of the other reviewers. Clearly a considerable amount of thought and effort has gone into their formulation. I remain convinced that this is an important paper that should be published. Nature Communications is a high-impact journal and accordingly these results will be widely disseminated and scrutinised. We must make sure that all details have been attended to appropriately.

The added introductory information is welcome and helps set the stage for the reader. However, I think some pruning of these paragraphs could still be performed- the purpose is to outline the narrative- the technical details are already in the methods section.

The additions and reorganisation of the supplementary mass univariate analyses are welcomed and generally strengthen the paper. That said, I would like to see a little more attention given to weaving these supplementary results into the discussion. The intention was not just to satisfy those of the reviewer; whether implicitly or explicitly, this work challenges the senior author's important original paper from 2003. I agree with the authors' viewpoints at ~line 395.

I thank the authors for providing clarity on their implementation of FIX and will look forward to reading the paper on the comparison with CompCor in the coming months. But, I still question the global signal normalisation approach at the first level and the statement that it generally reduces the impact of volume-to-volume global signal changes. What is important is whether the global signals are correlated with the task. If there is a relationship between the task and estimates of global signals, then spurious activations and deactivations can be interpreted within the data (e.g. https://www.fil.ion.ucl.ac.uk/spm/course/slides05-usa/pdf/Lab_D_GlobalScal.pdf) This is particularly likely when there are large and distributed clusters of task-related brain activity, such as is found in this work. The deactivations reported here are considerably more widespread in their anatomical location and may (at least in part) relate to global scaling. I would like to know: Does the global signal correlate with the task regressors in the heat pain experiments and if so, what is the effect of not performing global scaling on the widespread deactivations?

I am still not wholly satisfied by the NPS analyses. I understand that the 'beta * weight vector' approach has been performed previously (e.g. <https://www.ncbi.nlm.nih.gov/pmc/articles/PMC6248115/>), but this is not the same as a multivariate analysis. I note that the other two reviewers made very similar recommendations to my own, that the most appropriate course of action would be to perform a multivariate regression analysis. I do not agree with the author's assertion, however often repeated, that multivariate regression is a different question, or beyond the scope of the paper. The GLM mass univariate analysis performed is a regression analysis where one variable predicts the values of another. The comparable multivariate analysis is regression. The justification for not doing so, the 'potential introduction of bias' argument is a false premise. All three reviewers have requested multivariate regression, not ad nauseam testing, as is suggested.

The authors have also not responded to my question regarding the linearity of NPS co-efficients. The NEJM 2013 paper of Wager et al. states that the within-subject analyses of increasing perceived pain (Experiments 1 and 2) did not show linear increases in NPS score associated with more pain perceived at higher temperatures. There is the suggestion (Figure 1B) that the multivariate regression approach adopted by Wager and Atlas could predict pain intensity scores within an individual, but the regression slopes differ between individuals. In this work, the authors have sought to investigate linear relationships between brain activity and subjective report. This could be, but is not necessarily the same as a monotonic increase, as is suggested in the discussion. The authors should have

chosen to impose linear constraints. Given the range of scores, I do not have an issue with considering VAS as a linear scale, but whether that assumption holds for the NPS is less clear to me.

I think the authors have two choices here. If they wish to make claims about improved sensitivity to detect spatially distributed effects, then they must perform a multivariate regression analysis. The quicker and easier, but less optimal approach is to document the limitations of their current method in their discussion. I do not think that the current use of the NPS can be assumed to offer improved sensitivity as it is being used to perform decoding outside the parameters for which it was originally trained. Accordingly I think that the lines in the discussion (~ lines 370) suggesting this should be amended.

I note the new commentary at the end of the discussion regarding degeneracy/redundancy; concepts which are more deeply explored in the senior author's recent TICS paper. I do wonder whether perhaps these terms need to be defined for the wider readership of Nature Communications. Without wishing to overstep the mark as a reviewer, I agree that this is likely contributor to these findings. Melzack's original concept of an 'individualised body-self neuromatrix' based on shared biological hardware, but consistently shaped by one's life experiences, seems particularly relevant here. The lesion studies suggest that there are multiple overlapping systems/networks that can lead to a conscious pain experience despite damage in some of those networks. We already accept concepts of representational changes in persistent pain disorders- perhaps the relative contribution of each network differs more in healthy individuals than has been considered to date?

A few minor additional points-

-the reference to the default mode network in the results section (line ~211) feels like an aside and is misplaced here- other interpretations of their findings are not found in this section.

-The data look very spatially smooth. I note that a FWHM of 5mm is stated, but the input parameters for SUSAN are half-width in mm- so the equivalent FWHM is more than double that. Please check and amend if required.

-NPS details: It is not my real area of expertise, but if my memory serves correctly the cross product multiplication of two vectors leads to the production of a third, orthogonalized vector, rather than a scalar quantity. Please check and amend if required.

-Figure 6- a bit more description in the figure legend would be helpful- the purpose of the grey bars and the two different shaded data points should be added.

Reviewer #2 (Remarks to the Author):

Thanks for the authors' detailed rebuttals to my previous comments. I greatly appreciate the authors' efforts to revise their manuscript, which includes additional analysis on head motion and more discussions on degenerate neural mechanisms of pain, etc. However, it seems that some of my concerns were not resolved yet clearly. I wrote comments to some of the authors' responses below.

First, re: Univariate vs. Multivariate, I agree that developing a truly stable and reliable brain signature is a never-ending quest and requires substantial and continuous efforts of prospective testing. However, I think the authors' response seems to be a bit different from what I meant.

What is the rationale behind the use of NPS in this study? I thought, the authors used NPS to argue that it was impossible to find neural correlates of subjective pain intensity even though they used a multivariate-based approach. ("Thus, even when using a multivariate-based approach, known to be more sensitive than a massive univariate approach, brain activation supporting individual differences in reported pain was not identified.", p.17).

However, the fact that the NPS score was not correlated with subjective pain intensity does not preclude the possibility that we can detect brain activity patterns related to subjective pain intensity, because the NPS was mainly designed to predict the within-individual variation of pain ratings, not the between-individual variability. Thus, if the authors want to argue that the multivariate approach could not detect brain activity related to subjective pain intensity, they should try multivariate regression modeling using their own fMRI data and pain ratings, not the NPS which does not fit with this analysis.

Also, all neuroimaging findings suffer from replication crisis regardless of whether we used a univariate or multivariate approach. I did not require the authors to develop a robust biomarker that can be generalized across multiple datasets; just wanted to check whether the authors' argument works in a more appropriate framework. As I wrote in my previous comments, I don't want to complicate this paper which is already great as it is by adding new analyses. However, the descriptions about NPS and multivariate approach seem a bit misleading to me, and I think the authors should not preclude the possibility that the current null findings could be attributed to the well-known low sensitivity of univariate analysis, which is also closely related to the degenerate characteristics of neural representations of pain (Coghill 2020 TINS).

Second, re: rating bias, as the authors mentioned, there is a trend observed in 49°C-descending discrimination analysis. However, the whole fMRI analyses were for examining the neural representation of "reported pain intensity", not the unpleasantness. Therefore, the significance of the unpleasantness discrimination threshold in the 49°C-descending paradigm does not seem to directly support the absence of intensity rating bias. The lack of significance in 49°C is particularly important because the main fMRI analyses were done with 48°C heat stimulation, which is very close to 49°C. Not all the results with $p > 0.05$ are worthless, but considering the huge importance of this result for the current study (even appear in Abstract), I suggest the lack of statistical significance, at least, be noted and discussed in the main manuscript.

Third, re: spurious class, unfortunately, I still could not get what means the "spurious class". I'm wondering whether the authors could elaborate more on the definition of "spurious class", e.g., what is "true" population subgroup? I believe the authors defined the "spurious class" in a systematic and quantitative way, not by subjective or post-hoc decision. Is there any reference paper for determining this? This will help the possible readers who are not familiar with this type of analysis.

Reviewer #3 (Remarks to the Author):

The authors have addressed all of my comments from the initial review. This revised manuscript is an outstanding study.

January 21, 2022

NCOMMS-21-02776A

Dissociation between individual differences in self-reported pain intensity and underlying brain activation

Dear reviewers,

Thank you for taking the time to review our manuscript, NCOMMS-21-02776A. We found your reviews very helpful in pointing out issues to correct and refine, and we have revised the manuscript according to your comments.

Below we list the changes we made to the revised manuscript and respond to your concerns. These changes are highlighted in yellow in the manuscript. Your comments are written in italics and our detailed answers in regular font.

An important addition to the main manuscript is a multivariate LASSO-PCR analysis, which results further supported our main findings; this analysis did not identify a relationship between pain intensity ratings and brain activation in response to noxious heat stimuli.

While addressing your concerns, we were very careful to keep the focus of the paper on the relationship between individually perceived pain and underlying brain mechanisms. The dissociation between the two highlighted in this paper challenges long-standing scientific ideas based on single unit electrophysiology and neuroimaging. Accordingly, we seek to keep the emphasis of this manuscript on this important and novel finding.

Sincerely,

Marie-Eve Hoeppli, Ph.D.
Post-Doctoral Fellow

Robert C. Coghill, Ph.D.
Professor of Pediatrics
Director, Pediatric Pain Research Center

REVIEWER COMMENTS

Reviewer #1 (Remarks to the Author):

Response to resubmission- Hoeppli et al

I thank the authors for their detailed responses to my comments and those of the other reviewers. Clearly a considerable amount of thought and effort has gone into their formulation. I remain convinced that this is an important paper that should be published. Nature Communications is a high-impact journal and accordingly these results will be widely disseminated and scrutinised. We must make sure that all details have been attended to appropriately.

Authors' response (AR): We thank the reviewer for his/her appreciation of our work and for his/her support to increase the quality of our manuscript for publication.

The added introductory information is welcome and helps set the stage for the reader. However, I think some pruning of these paragraphs could still be performed- the purpose is to outline the narrative- the technical details are already in the methods section.

AR: As suggested by the reviewer, we removed the technical details from the introduction.

The additions and reorganisation of the supplementary mass univariate analyses are welcomed and generally strengthen the paper. That said, I would like to see a little more attention given to weaving these supplementary results into the discussion. The intention was not just to satisfy those of the reviewer; whether implicitly or explicitly, this work challenges the senior author's important original paper from 2003. I agree with the authors' viewpoints at ~line 395.

AR: we thank the reviewer for his/her suggestion to add more discussion on the supplementary analyses, which strengthen our claim that interindividual differences in pain are not associated with differences in brain activation. The added comments are highlighted in the discussion.

I thank the authors for providing clarity on their implementation of FIX and will look forward to reading the paper on the comparison with CompCor in the coming months.

AR: We thank the reviewer for his/her continued interest in our work.

But, I still question the global signal normalisation approach at the first level and the statement that it generally reduces the impact of volume-to-volume global signal changes. What is important is whether the global signals are correlated with the task. If there is a relationship between the task and estimates of global signals, then spurious activations and deactivations can be interpreted within the data (e.g. https://www.fil.ion.ucl.ac.uk/spm/course/slides05-usa/pdf/Lab_D_GlobalScal.pdf) This is particularly likely when there are large and distributed clusters of task-related brain activity, such as is found in this work. The deactivations reported

here are considerably more widespread in their anatomical location and may (at least in part) relate to global scaling. I would like to know: Does the global signal correlate with the task regressors in the heat pain experiments and if so, what is the effect of not performing global scaling on the widespread deactivations?

We thank the reviewer for his/her interest in this important manner. We extracted global signal and correlated the time course of global signal with the stimulus time course for every series for each individual participant. The individual correlations between whole-brain global signal and stimulus time courses showed a wide range of r-values (-0.54 to 0.65). Averaged r-values were in the no to low correlation range (overall average \pm sd = -0.001 ± 0.21), indicating no systematic relationship between the stimulus time course and global signal. Importantly, r-values did not seem to depend on the pain sensitivity classes and/or the heat series. The scatterplot below shows the lack of relationship between pain intensity rating of 48°C stimuli and r-values for each heat series.

As recommended, we compared data without intensity normalization with the original analysis using intensity normalization. Consistent with the original normalized analysis, no relationship was identified between pain sensitivity and brain activation.

Examination of the main effect of stimulation showed that global normalization resulted in a reduction of noise and an increase in signal detection in response to high intensity heat stimuli, as shown in the figure below. Marked improvements were obtained after global normalization, such that activations became clearly localized and clearly resemble those obtained in the majority of studies of noxious heat. Conversely, the deactivations in the data with global normalization largely occurred within the same brain regions as those that were identified in data without global normalization. Thus, global normalization appears to minimize potentially spurious signal changes in the grey matter by reducing noise. White matter signals are differentially impacted, however.

The important point to emphasize from these analyses is that global signal correlations do not appear to vary as a function of pain sensitivity. Thus, by minimizing this source of noise, our ability to detect a relationship between pain sensitivity and brain activation would be enhanced rather than diminished.

The topic of global signal change is one of substantial interest, and much like the comparison between data cleaning methodologies, would be best addressed in a separate publication focusing specifically on them and their impact on interpretation of data.

*I am still not wholly satisfied by the NPS analyses. I understand that the 'beta * weight vector' approach has been performed previously (e.g. <https://www.ncbi.nlm.nih.gov/pmc/articles/PMC6248115/>), but this is not the same as a multivariate analysis. I note that the other two reviewers made very similar recommendations to my own, that the most appropriate course of action would be to perform a multivariate regression analysis. I do not agree with the author's assertion, however often repeated, that multivariate regression is a different question, or beyond the scope of the paper. The GLM mass univariate analysis performed is a regression analysis where one variable predicts the values of another. The comparable multivariate analysis is regression. The justification for not doing so, the 'potential introduction of bias' argument is a false premise. All three reviewers have requested multivariate regression, not ad nauseam testing, as is suggested.*

AR: As suggested by both reviewers, we have now performed a multivariate LASSO-PCR regression following Wager et al. (2011 and 2013) procedure that we trained on our dataset. We included a five-fold cross validation test that was repeated five times and a bootstrap test with 5000 samples. Results from this analysis showed no relationship of perceived pain intensity based on brain activation associated with high intensity heat stimuli. This result supports our main findings that there is no relationship between perception of pain intensity and brain activations associated with a highly noxious heat stimulus. Details of the applied method and result has been added and highlighted in the main text of the manuscript. A scatterplot showing the lack of relationship between predicted and observed pain intensity has been included in the supplementary figures (Supplementary Figure 4).

The authors have also not responded to my question regarding the linearity of NPS co-efficients. The NEJM 2013 paper of Wager et al. states that the within-subject analyses of increasing

perceived pain (Experiments 1 and 2) did not show linear increases in NPS score associated with more pain perceived at higher temperatures. There is the suggestion (Figure 1B) that the multivariate regression approach adopted by Wager and Atlas could predict pain intensity scores within an individual, but the regression slopes differ between individuals. In this work, the authors have sought to investigate linear relationships between brain activity and subjective report. This could be, but is not necessarily the same as a monotonic increase, as is suggested in the discussion. The authors should have chosen to impose linear constraints. Given the range of scores, I do not have an issue with considering VAS as a linear scale, but whether that assumption holds for the NPS is less clear to me.

AR: **Within** individuals, graded increases in noxious heat have long been known to be related to perceived pain intensity with a positively accelerating stimulus response function following the classic Steven's power law. However, **between** individual data from our 2003 paper suggests a largely linear relationship between brain activity and **individual** differences in perceived pain intensity. In Wager NEJM 2013, the relationship between the NPS expression and pain rating approximates a linear relationship (Figure 2B) within the noxious range. Thus, we tested for a linear relationship between individual differences in perceived pain intensity and brain activation in both the mass univariate analysis as well as the NPS analysis.

I think the authors have two choices here. If they wish to make claims about improved sensitivity to detect spatially distributed effects, then they must perform a multivariate regression analysis. The quicker and easier, but less optimal approach is to document the limitations of their current method in their discussion. I do not think that the current use of the NPS can be assumed to offer improved sensitivity as it is being used to perform decoding outside the parameters for which it was originally trained. Accordingly I think that the lines in the discussion (~ lines 370) suggesting this should be amended.

AR: We have performed the multivariate regression and have changed the discussion to not state that the NPS has greater sensitivity.

I note the new commentary at the end of the discussion regarding degeneracy/redundancy; concepts which are more deeply explored in the senior author's recent TICS paper. I do wonder whether perhaps these terms need to be defined for the wider readership of Nature Communications. Without wishing to overstep the mark as a reviewer, I agree that this is likely contributor to these findings. Melzack's original concept of an 'individualised body-self neuromatrix' based on shared biological hardware, but consistently shaped by one's life experiences, seems particularly relevant here. The lesion studies suggest that there are multiple overlapping systems/networks that can lead to a conscious pain experience despite damage in some of those networks. We already accept concepts of representational changes in persistent pain disorders- perhaps the relative contribution of each network differs more in healthy individuals than has been considered to date?

AR: We thank the reviewer for his/her suggestion and added the following statement to the discussion to clarify the background of the multiple realizability concept: "Perhaps the

resilience suggested by lesion studies^{48,49} may reflect the ability of pain to be instantiated by multiple distinct systems, and this multiple realizability may be more important across healthy individuals than previously thought.

A few minor additional points-

-the reference to the default mode network in the results section (line ~211) feels like an aside and is misplaced here- other interpretations of their findings are not found in this section.

AR: We thank the reviewer for this keen observation. We removed this interpretation from the results.

-The data look very spatially smooth. I note that a FWHM of 5mm is stated, but the input parameters for SUSAN are half-width in mm- so the equivalent FWHM is more than double that. Please check and amend if required.

AR: We appreciate the reviewer's attention to details. The FWHM that was used was indeed 5mm. The spatial size for smoothing used to run the SUSAN algorithm was automatically calculated by the FSL FEAT tool.

-NPS details: It is not my real area of expertise, but if my memory serves correctly the cross product multiplication of two vectors leads to the production of a third, orthogonalized vector, rather than a scalar quantity. Please check and amend if required.

AR: We thank the reviewer for catching this error. The dot product was used to obtain the scalar value, not the cross product. We corrected the methods of the NPS accordingly.

-Figure 6- a bit more description in the figure legend would be helpful- the purpose of the grey bars and the two different shaded data points should be added.

AR: We thank the reviewer for this comment. We further explained this complicated graph by modifying this figure legend as follow: The average NPS expression, represented by the bars, is higher in high intensity heat stimulation compared to low intensity heat stimulation. In addition, the violin plots, which represent the distribution of individual NPS expression in each intensity, show a greater density of individuals with higher NPS expression in the high intensity heat stimulation than in the low intensity heat stimulation. Finally, the error bars represent one standard error of the mean. *** represents p values below 0.001.

Reviewer #2 (Remarks to the Author):

Thanks for the authors' detailed rebuttals to my previous comments. I greatly appreciate the authors' efforts to revise their manuscript, which includes additional analysis on head motion and more discussions on degenerate neural mechanisms of pain, etc. However, it seems that

some of my concerns were not resolved yet clearly. I wrote comments to some of the authors' responses below.

First, re: Univariate vs. Multivariate, I agree that developing a truly stable and reliable brain signature is a never-ending quest and requires substantial and continuous efforts of prospective testing. However, I think the authors' response seems to be a bit different from what I meant.

What is the rationale behind the use of NPS in this study? I thought, the authors used NPS to argue that it was impossible to find neural correlates of subjective pain intensity even though they used a multivariate-based approach. ("Thus, even when using a multivariate-based approach, known to be more sensitive than a massive univariate approach, brain activation supporting individual differences in reported pain was not identified.", p.17).

However, the fact that the NPS score was not correlated with subjective pain intensity does not preclude the possibility that we can detect brain activity patterns related to subjective pain intensity, because the NPS was mainly designed to predict the within-individual variation of pain ratings, not the between-individual variability. Thus, if the authors want to argue that the multivariate approach could not detect brain activity related to subjective pain intensity, they should try multivariate regression modeling using their own fMRI data and pain ratings, not the NPS which does not fit with this analysis.

Also, all neuroimaging findings suffer from replication crisis regardless of whether we used a univariate or multivariate approach. I did not require the authors to develop a robust biomarker that can be generalized across multiple datasets; just wanted to check whether the authors' argument works in a more appropriate framework. As I wrote in my previous comments, I don't want to complicate this paper which is already great as it is by adding new analyses. However, the descriptions about NPS and multivariate approach seem a bit misleading to me, and I think the authors should not preclude the possibility that the current null findings could be attributed to the well-known low sensitivity of univariate analysis, which is also closely related to the degenerate characteristics of neural representations of pain (Coghill 2020 TINS).

AR: We appreciate both reviewers' concerns about the application of the NPS. We have now applied a multivariate LASSO-PCR model to our data, training it on our dataset. (See response to reviewer 1 above.) The results of this model further support our finding of a lack of relationship between perceived pain intensity and brain activation associated with high intensity heat stimuli, by failing to predict pain intensity based on brain activation associated with these stimuli. We reported the methods and results of this new analysis in the main text of the manuscript and added a scatterplot illustrating the lack of relationship between predicted and observed pain intensity in Supplementary Figure 4.

Second, re: rating bias, as the authors mentioned, there is a trend observed in 49°C-descending discrimination analysis. However, the whole fMRI analyses were for examining the neural representation of "reported pain intensity", not the unpleasantness. Therefore, the significance of the unpleasantness discrimination threshold in the 49°C-descending paradigm does not seem

to directly support the absence of intensity rating bias. The lack of significance in 49°C is particularly important because the main fMRI analyses were done with 48°C heat stimulation, which is very close to 49°C. Not all the results with $p > 0.05$ are worthless, but considering the huge importance of this result for the current study (even appear in Abstract), I suggest the lack of statistical significance, at least, be noted and discussed in the main manuscript.

AR: We thank the reviewer for this comment. We have now clarified this finding in the results section (“Kruskal-Wallis tests performed on the 49°C-descending discrimination thresholds revealed no significant difference between classes in intensity ratings. They only revealed a significant difference between classes in unpleasantness ratings”) and are discussing it further in the discussion section of the main manuscript: “This lack of statistically significant results in the pain intensity ratings could be driven by the smaller difference in temperature decrease needed to achieve perceptive change in pain intensity compared to pain unpleasantness between the Low and High Pain Sensitivity classes. These results were also supported by a correlation analysis performed to ensure that our mixture model approach did not affect the results.”

Third, re: spurious class, unfortunately, I still could not get what means the “spurious class”. I’m wondering whether the authors could elaborate more on the definition of “spurious class”, e.g., what is “true” population subgroup? I believe the authors defined the “spurious class” in a systematic and quantitative way, not by subjective or post-hoc decision. Is there any reference paper for determining this? This will help the possible readers who are not familiar with this type of analysis.

AR: We thank the reviewer for his comment on this advanced statistical approach. We now added the following definition of mixture models with spurious class to the methods of the main paper and included the appropriate reference: “Mixture models including spurious classes were defined as models, in which one of the classes contained 10% or less of the participants included in our dataset. In this case, the mixture solution was considered to be unlikely to reflect real sub-populations or classes of Pain Sensitivity and more likely to reflect an over-extraction of classes⁸⁴”. We hope that this will help clarify the criteria used for the selection of the mixture model.

Reviewer #3 (Remarks to the Author):

The authors have addressed all of my comments from the initial review. This revised manuscript is an outstanding study.

AR: We thank the reviewer for his enthusiastic support and involvement in the improvement of our manuscript.

REVIEWER COMMENTS

Reviewer #1 (Remarks to the Author):

NCOMMS-21-02776A: Dissociation between individual differences in self-reported pain intensity and underlying brain activation.

Dear authors,

Thank you again for your considered responses and additional work. I recognise how frustrating it is to be repeatedly asked to go back over a dataset to satisfy a reviewer, so thought I would make it clear early in my response this time that I am not asking for any more analysis and will make my recommendation that the paper should be published.

I think that the paper is strengthened by the additional multivariate analysis. I was initially tempted to ask for more info on how the LASSO-PCR analysis was performed, as the details on the Canlab web resource are exemplars that relate to within-subject analyses, such as were described in the original NPS papers. Importantly, however, I can see how the software may be modified to perform multivariate regression to examine individual differences as is required here. On balance I decided the paper is hefty enough as it is without more methods info. The finding that no relationship is identified from your data adds strength to the central tenet of the report that we can't discern a relationship between individual differences in subjectively reported responses to heat pain and BOLD responses.

Thank you for the extra info on the global signal intensity normalisation. It looks likely to me like some of the white matter activation you are seeing is due to this processing step; in situations where there is significant correlation between the global signal and the task, voxels that are not activated may be interpreted as spurious deactivations. This phenomenon has been described before- see Aguirre, G. K., Zarahn, E., & D'Esposito, M. (1998). The inferential impact of global signal covariates in functional neuroimaging analyses. *Neuroimage*, 8(3), 302-306. I recommend that you briefly consider this in your interpretation of white matter activity around line 308, but I also agree that we should not get too bogged down in considering this here.

(A related aside- 308 also has a typo- s(t)imulation?; also check 'fit/fitted' on line 85).

Thanks for tidying up the minor errors and improving the figure legends. I think we can reasonably expect that a lot of people will take notice of this report and so making it as accessible as it is robust is important.

As a side note, I have repeatedly included the 2003 report in lectures to my undergraduates on the use of fMRI in pain research. I will update those slides upon publication of this important work. Well done to the authorship team.

Reviewer #2 (Remarks to the Author):

The authors have provided very clear and thorough responses to my comments. I also appreciate the authors' efforts to perform multivariate regression modeling.

My last comment is that, there is no description about the hyperparameter selection for LASSO shrinkage. Was the lambda determined to minimize the MSE in the training set, or is there any other cost function used? Do the lambda values vary across the 5 folds? Did the authors perform nested cross-validation to mitigate overfitting in the training set? PCR-based multivariate modeling on fMRI data is very susceptible to the selection of PC numbers, so clarification on these issues is needed.

March 10, 2022

NCOMMS-21-02776B

Dissociation between individual differences in self-reported pain intensity and underlying brain activation

Dear reviewers,

Thank you for taking the time to review our manuscript, NCOMMS-21-02776B. We found your reviews very helpful in pointing out issues to correct and refine, and we have revised the manuscript according to your comments.

Below we list the changes we made to the revised manuscript and respond to your concerns. Your comments are written in italics and our detailed answers in regular font. The changes are highlighted in yellow in the manuscript.

Sincerely,

Marie-Eve Hoeppli, Ph.D.
Post-Doctoral Fellow

Robert C. Coghill, Ph.D.
Professor of Pediatrics
Director, Pediatric Pain Research Center

REVIEWERS' COMMENTS

Reviewer #1 (Remarks to the Author):

NCOMMS-21-02776A: Dissociation between individual differences in self-reported pain intensity and underlying brain activation.

Dear authors,

Thank you again for your considered responses and additional work. I recognise how frustrating it is to be repeatedly asked to go back over a dataset to satisfy a reviewer, so thought I would make it clear early in my response this time that I am not asking for any more analysis and will make my recommendation that the paper should be published.

Authors' Response (AR): We sincerely thank the reviewer for his/her support to publish our manuscript.

I think that the paper is strengthened by the additional multivariate analysis. I was initially tempted to ask for more info on how the LASSO-PCR analysis was performed, as the details on the Canlab web resource are exemplars that relate to within-subject analyses, such as were described in the original NPS papers. Importantly, however, I can see how the software may be modified to perform multivariate regression to examine individual differences as is required here. On balance I decided the paper is hefty enough as it is without more methods info. The finding that no relationship is identified from your data adds strength to the central tenet of the report that we can't discern a relationship between individual differences in subjectively reported responses to heat pain and BOLD responses.

AR: We thank the reviewer for his/her reasoning on the additional results and agree that the addition of the multivariate analysis further reinforces the main claim of this paper.

*Thank you for the extra info on the global signal intensity normalisation. It looks likely to me like some of the white matter activation you are seeing is due to this processing step; in situations where there is significant correlation between the global signal and the task, voxels that are not activated may be interpreted as spurious deactivations. This phenomenon has been described before- see Aguirre, G. K., Zarahn, E., & D'Esposito, M. (1998). The inferential impact of global signal covariates in functional neuroimaging analyses. *Neuroimage*, 8(3), 302-306. I recommend that you briefly consider this in your interpretation of white matter activity around line 308, but I also agree that we should not get too bogged down in considering this here.*

AR: We thank the reviewer for his/her comment and have now added the following explanation to the main manuscript: "It has been previously shown that inclusion of global signal confounds, which are correlated to the experimental paradigm, i.e. stimulation, might affect the relationship between the paradigm, and the resulting brain activations³⁷. However, given that

the correlation of global signal confounds and our stimulation is close to null across participants, this effect is likely to be small and individualized.”

(A related aside- 308 also has a typo- s(t)imulation?; also check ‘fit/fitted’ on line 85). Thanks for tidying up the minor errors and improving the figure legends. I think we can reasonably expect that a lot of people will take notice of this report and so making it as accessible as it is robust is important.

AR: We thank the reviewer for his/her keen eye. We have now corrected our typos and grammatical errors.

As a side note, I have repeatedly included the 2003 report in lectures to my undergraduates on the use of fMRI in pain research. I will update those slides upon publication of this important work. Well done to the authorship team.

AR: We thank the reviewer for including our previous and current publications in his/her lectures.

Reviewer #2 (Remarks to the Author):

The authors have provided very clear and thorough responses to my comments. I also appreciate the authors’ efforts to perform multivariate regression modeling.

AR: We thank the reviewer for his/her appreciation of our revisions.

My last comment is that, there is no description about the hyperparameter selection for LASSO shrinkage. Was the lambda determined to minimize the MSE in the training set, or is there any other cost function used? Do the lambda values vary across the 5 folds? Did the authors perform nested cross-validation to mitigate overfitting in the training set? PCR-based multivariate modeling on fMRI data is very susceptible to the selection of PC numbers, so clarification on these issues is needed.

AR: We can confirm to the reviewer that we used nested cross-validation and that the lambda was estimated to minimize the MSE. Lambda values across folds ranged from 0.15 to 0.34 (average: 0.25).

We have added these details to the description of the multivariate analysis in the Methods section of the main manuscript. We would also like to draw the attention of the reviewer on the fact that, given our lack of significant results in this analysis, the risk of overfitting the data is low.